

# Outer Van Allen belt trapped and precipitating electron flux responses to two interplanetary magnetic clouds of opposite polarity

Harriet George[1], Emilia Kilpua[1], Adnane Osmane[1], Timo Asikainen[2], Milla M. H. Kalliokoski[1], Craig J. Rodger[3], Stepan Dubyagin[4], and Minna Palmroth[1]

[1]Department of Physics, University of Helsinki, Helsinki, Finland
[2]Department of Physics, University of Oulu, Oulu, Finland
[3]Department of Physics, University of Otago, Dunedin, New Zealand
[4]Finnish Meteorological Institute, Helsinki, Finland

**Correspondence:** Harriet George (harriet.george@helsinki.fi)

**Abstract.** Recently, it has been established that interplanetary coronal mass ejections (ICMEs) can dramatically affect both trapped electron fluxes in the outer radiation belt and precipitating electron fluxes lost from the belt into the atmosphere. Precipitating electron flux and energy can vary over a range of timescales during these events. These variations depend on the initial energy and location of the electron population, as well as the ICME characteristics and structures. One important factor controlling electron dynamics is the magnetic field orientation within the ejecta that is an integral part of the ICME. In this study, we examine Van Allen Probes (RBSP) and Polar Orbiting Environmental Satellites (POES) data to explore trapped and precipitating electron fluxes during two ICMEs. The ejecta in the selected ICMEs have magnetic cloud characteristics that exhibit opposite sense of rotation of the north-south magnetic field component ($B_Z$). RBSP data are used to study trapped electron fluxes in situ, while POES data are used for electron fluxes precipitating into the upper atmosphere. The trapped and precipitating electron fluxes are qualitatively analysed to understand their variation in relation to each other and to magnetic cloud rotation during these events. Inner magnetospheric wave activity was also estimated using RBSP and Geostationary Operational Environmental Satellite (GOES) data. In each event, the largest changes in the location and magnitude of both the trapped and precipitating electron fluxes occurred during the southward portion of the magnetic cloud. Significant changes also occurred during the end of the sheath and at the sheath-cloud boundary for the cloud with south to north magnetic field rotation, while the ICME with north to south rotation had significant changes at the end boundary of the cloud. The sense of rotation of $B_Z$ and its profile also clearly affects the coherence of the trapped/precipitating flux changes, timing of variations with respect to the ICME structures, and flux magnitude of different electron populations. The differing electron responses could therefore imply partly different dominant acceleration mechanisms acting on the outer radiation belt electron populations as a result of opposite magnetic cloud rotation.

## 1 Introduction

The Van Allen radiation belts are highly dynamic regions of charged particles trapped in the Earths geomagnetic field (Van Allen, 1981). The traditional picture of the belts consists of two toroidal regions of energetic particles that are separated





by a relatively empty slot region. One of these two regions is the inner belt, which is located at $\sim 1.2R_E < r < 2R_E$ and is dominated by high-energy protons, while the outer belt is located at $\sim 3R_E < r < 10R_E$ and is dominated by energetic

electrons. The structure of the outer belt is highly dynamic, and electron fluxes can vary drastically with timescales ranging from seconds to months (e.g., Baker et al., 2018; Reeves et al., 2013). The radiation belts encompass regions that various satellites, including communications and navigation satellites, pass through. Exposure to high levels of radiation can damage these satellites, especially the sensitive electronics on-board, thus shortening the lifespan and reducing the functionality of the satellites. Another key societal interest related to radiation belt electron dynamics is the precipitation of energetic electrons

from the belts to the upper atmosphere (e.g., Rodger et al., 2010; Verronen et al., 2011). Precipitating electrons cause pulsating aurora (e.g., Partamies et al., 2017; Grandin et al., 2017) and they can affect chemical composition and dynamics of the middle atmosphere, and therefore are important to take into account in climate models (e.g., Maliniemi et al., 2013; Salminen et al., 2019; Seppälä et al., 2014). It is thus highly important to be able to understand radiation belt trapped electron fluxes as well as the precipitating electron fluxes lost from the belts into the upper atmosphere.

Recently, outer radiation belt electrons have been divided into source, seed, core and ultrarelativistic populations depending on their energy. Source electrons are tens of kiloelectron volts (keV) (e.g., Jaynes et al., 2015), seed electrons are hundreds of keVs (e.g., Jaynes et al., 2015), core electrons range from about 500 keV to 1-2 megaelectron volts (MeV) (e.g., Boyd et al., 2016), and ultrarelativistic electrons have energies from several MeVs upwards (e.g., Boyd et al., 2016). Changes in electron flux in the outer radiation belt may be due to either temporary adiabatic variations (Li et al., 1997; Tu and Li, 2011), or genuine

loss or gain processes. These processes affect different electron populations in different ways, and are modulated by various geospace conditions, such as geomagnetic storms.

Real loss or gain processes result in a change in the flux of a given population that persists after storm recovery. In the outer radiation belt, the two main loss processes that genuinely displace particles outside the belt are magnetopause shadowing and precipitation into the upper atmosphere. Magnetopause shadowing occurs when electron drift trajectories cross the

magnetopause, resulting in permanent loss from the belt to the solar wind (e.g., Turner et al., 2014). This requires that the magnetopause is displaced significantly Earthward, which occurs through compression due to enhanced solar wind dynamic pressure and/or erosion due to a strongly southward interplanetary magnetic field (Shue et al., 1998). Shadowing losses can be enhanced by outward radial diffusion of electrons (e.g., Mann et al., 2016) and the so-called Dst effect (e.g., Kim and Chan, 1997; Li et al., 1997), i.e., the adiabatic expansion of electron drift paths and consequent adiabatic electron deceleration when

enhanced ring current weakens the Earth's magnetic field. Electron precipitation occurs when particles enter the loss cone, and is typically caused by wave-particle interactions or major reconfiguration of the magnetotail (e.g., Sivadas et al., 2017). It has been suggested that the primary loss mechanism of radiation belt particles during geomagnetic storms is precipitation into the upper atmosphere (e.g., Clilverd et al., 2006, 2007). When this occurs, the electrons collide with atmospheric constituents and are thus permanently lost from the radiation belt.

A number of plasma waves exist in the outer belt region of the inner magnetosphere that interact with the various trapped electron populations in this region (e.g., Baker et al., 2018). Electromagnetic ion cyclotron (EMIC) waves, for example, have strong linear resonance with > 2 MeV electrons (e.g., Meredith et al., 2003; Usanova et al., 2014). EMIC waves also interact





non-resonantly with electrons of energies as low as a few hundred keV, and these low-energy interactions may be elevated
in geomagnetically active conditions (e.g., Blum et al., 2019; Hendry et al., 2017). Whistler mode chorus waves effectively

scatter lower energy electrons (few tens of keV) (e.g., Lam et al., 2010), and large-amplitude chorus waves can also lead to
rapid scattering of MeV electrons in microburst events (e.g., Douma et al., 2017; Osmane et al., 2016). Chorus waves arise
from source electron injections during substorms, and are thought to be the primary cause of progressive acceleration of lower
energy seed electrons to relativistic energies (e.g., Jaynes et al., 2015; Rodger et al., 2016). Acceleration of seed electrons can
also occur via inward radial diffusion by ultra-low frequency (ULF) Pc4-Pc5 waves (e.g., Jaynes et al., 2018; Mann et al.,

2013). Plasmaspheric hiss is the key precipitating wave mode in the plasmasphere, and timescales for interaction with this
wave mode range from a few hours for source electron energies to several days for relativistic electrons (Kavanagh et al.,
2018; Li et al., 2015; Selesnick et al., 2003). These various waves are present at different times and locations and can cause
both acceleration and pitch angle scattering of electrons (e.g., Rodger et al., 2019). Precipitation into the upper atmosphere
occurs if pitch angles are changed in such a way that electrons enter the loss cone. Interactions can lead to either enhancements

or depletions of the electron population, depending on the energy of the population and wave properties (e.g., Rodger et al.,
2007). The variation of trapped flux of electrons in the outer radiation belt and the subsequent precipitation into the upper
atmosphere is therefore extremely complex due to the wide range of processes acting on the various electron populations. The
radiation belt system is constantly influenced by changing solar wind conditions impinging on the Earth's magnetosphere,
with the most drastic variations occurring during large-scale heliospheric structures. In this work, we focus on the impact

of interplanetary coronal mass ejections (ICMEs, e.g., Kilpua et al., 2017) on the radiation belt system. An ICME is the
interplanetary counterpart of a sudden ejection of magnetised plasma from the Sun. The integral component of the ICME
is the driver ejecta that regularly features a magnetic flux rope or a "magnetic cloud" configuration with smoothly changing
magnetic field direction and enhanced magnetic field magnitude (e.g., Burlaga et al., 1981). When an ICME is sufficiently
faster than the preceding solar wind, a leading shock and a sheath region forms ahead of the ejecta. Sheaths typically feature

a turbulent magnetic field and high dynamic pressure due to their compressive nature. The sheath and ejecta of ICMEs are
known to have significantly different impacts on electrons fluxes within the outer radiation belt (Kalliokoski et al., 2019), and
magnetic clouds are a particularly geoeffective subset of ICME ejecta. One feature of magnetic clouds that strongly affects their
geoeffectiveness is the magnetic polarity (e.g., Bothmer and Schwenn, 1998; Huttunen et al., 2005). Southward magnetic fields
undergo effective magnetic reconnection at the dayside magnetopause and subsequently enable efficient transfer of solar wind

energy and plasma to the magnetosphere, inducing large changes in the radiation belt system. Northward magnetic fields have
less effective reconnection, and therefore have a less pronounced impact on radiation belt dynamics than southward magnetic
clouds. Some magnetic clouds have southward fields in their leading part that rotate to northward orientation by the trailing
edge (South-North or SN type cloud), so turbulent sheath fields are close to southward fields in the cloud. Other magnetic
clouds exhibit opposite rotation (North-South or NS type cloud). NS type clouds may enhance geoeffectivity following the end

of the ICME due to interactions between the southward magnetic field and the trailing solar wind (e.g., Fenrich and Luhmann,
1998; Kilpua et al., 2012).





Due to the complexity of the radiation belt response, the impact of ICMEs on outer radiation belt electrons is not yet fully understood. ICMEs can cause enhancements, depletions, or no change in electron flux in a given population depending on the ICME characteristics and initial radiation belt conditions (e.g., Kilpua et al., 2015a; Reeves et al., 2003; Turner et al., 2019).

ICMEs have also been observed to trigger strong responses in precipitating flux over a range of locations and time scales (e.g., Clilverd et al., 2007). However, the relationship between electron precipitation into the upper atmosphere and enhancement or depletion of trapped electron flux in the outer radiation belt remains an open research area.

In this study, we compare data from Polar Orbiting Environmental Satellites (POES) and Van Allen Probes (RBSP) for two case studies of ICMEs near the maximum of Solar Cycle 24 with SN (31/12/2015) and NS (27/06/2013) magnetic cloud

rotations. The Van Allen Probes measure trapped electron fluxes within the radiation belts while POES provides a measure of the electron fluxes precipitating into the upper atmosphere. We use these data to qualitatively analyse the relationship between outer radiation belt trapped electron fluxes, from source to ultrarelativistic energies, and precipitating fluxes during these two ICMEs. Of particular interest is the electron response to different magnetic field orientations within each event, and also to opposite magnetic cloud rotations in the two case studies.

## 2 Data and Methods


The two ICME events that were evaluated in this study impacted the Earth on 26 June 2013 and 30 December 2015. The start and end times of these ICMEs were taken from the Cane and Richardson online database (http://www.srl.caltech.edu/ACE/ASC/DATA/level3/icmetable2.htm; Richardson and Cane, 2010). As discussed in the Introduction, these two ICMEs were selected due to their clear magnetic cloud configuration and opposite rotation of magnetic polarity (SN and NS, respectively).

Additionally, both ICME drove a leading shock, enabling comparison of the impacts of ICME sheath and ejecta on electron dynamics, and comprehensive datasets from various satellites were available for the duration of the two events.

Solar wind data are obtained from the Wind spacecraft. The Magnetic Fields Investigation (MFI) is used for magnetic field data, and solar wind speed and pressure are measured by the Solar Wind Experiment (SWE). The Disturbance Storm Time (Dst) data are retrieved from WDC Kyoto in one hour time resolution. The AL index, a measure of the strongest current intensity of

the westward auroral electrojet, is taken from the OMNI database in one minute time resolution. The magnetopause position is calculated from the Shue et al. (1998) model, using the five minute means of the $Z$-component of the magnetic field ($B_z$) and dynamic pressure ($P_{Dyn}$). The plasmapause location is calculated from maximum AE values in one hour time intervals, as detailed in O'Brien and Moldwin (2003).

We also investigate wave activity in the inner magnetosphere. The chorus and hiss wave activity is investigated in magnetic

spectral intensity spectrograms compiled using RBSP Magnetic Field Instrument Suite and Integrated Science (EMFISIS) (Kletzing et al., 2013) magnetometer Level 2 data from the EMFISIS website (https://emfisis.physics.uiowa.edu/data/index). The electron cyclotron frequencies at the equator ($f_{ce,eq}$) were calculated based on the Tsyganenko and Sitnov geomagnetic field model (TS04D) (Tsyganenko and Sitnov, 2005). Usually lower-band whistler mode chorus waves occur between $0.1f_{ce,eq} < f < 0.5f_{ce,eq}$ and the upper-band between $0.5f_{ce,eq} < f < 1.0f_{ce,eq}$ (e.g., Burtis and Helliwell, 1969; Koons



and Roeder, 1990), but patches of chorus waves that continue to lower frequencies have been observed at higher latitudes
(*e.g.*, Cattell et al., 2015; Xiao et al., 2017). Plasmaspheric hiss occurs typically above 100 Hz and below $\sim 0.1 f_{ce,eq}$ in-
side the plasmasphere (e.g., Hartley et al., 2018). The wave power in the ULF Pc5 range (2.5 to 10 minutes or frequencies
from 2 to 7 mHz) and EMIC range (from 0.2 to 10 seconds or frequencies 0.1 – 5 Hz) were calculated using the geosta-
tionary GOES-13 and GOES-15 spacecraft magnetometer (Singer et al., 1996) 0.512–second magnetic field data obtained

through https://www.ngdc.noaa.gov/stp/satellite/goes/dataaccess.html. We first calculated wave power spectral density using
the wavelet analysis of the magnetic field magnitude and then defined wave power in the above mentioned Pc5 and EMIC
ranges.

The electron flux measurements were obtained from the Van Allen Probes, which are twin spacecraft (RBSP-A and RBSP-B)
on highly elliptical orbits through the radiation belts. Four energies were evaluated in this study: 32keV, 356.0keV, 1.066MeV,

and 3.4MeV. These energies correspond to examples of source, seed, core, and ultrarelativistic electron flux populations
respectively. The three lower energy channels were measured by the Magnetic Electron Ion Spectrometer (MagEIS: Blake
et al. (2013)), while ultrarelativistic electron flux was measured by Relativistic Electron Proton Telescope (REPT: Baker et al.
(2012)). Data from the REPT instrument of RBSP-B were unavailable on 30/12/2015, at the start of Event 1, and MagEIS
data from RBSP-A were unavailable on 30/6/2013, the final day included in the Event 2 analysis. The Level 2 spin-averaged

electron flux data, which has temporal resolution of $\sim 11$ s, were used in this study.

Data from the Medium Energy Proton and Electron Detector (MEPED) instrument of the Space Environment Monitor (SEM-
2) suite onboard the Polar Orbiting Environmental Satellites (POES) were used for the electron precipitation data. The POES
satellites used for Event 1 analysis were NOAA-15, NOAA-18, NOAA-19, METOP-01, and METOP-02, and Event 2 analysis
used POES data from all of these satellites as well as NOAA-16. POES are well suited for precipitation studies as they orbit the

polar regions at low altitudes, near the feet of the Earth's geomagnetic field lines where precipitation takes place (Rodger et al.,
2010b). The MEPED instrument contains two telescopes: $0°$ and $90°$ that are oriented almost radially outward from the Earth
and almost antiparallel to satellite velocity, respectively. Each telescope measures electron flux ($J_0$ and $J_{90}$, respectively). At
high latitudes, the $0°$ telescope measures precipitating particles and the $90°$ telescope measures trapped particles (Rodger et al.,
2010a). Both telescopes have a $30°$ angle of view and records electron flux in three energy channels: $> 30$ keV, $> 100$ keV, and

$> 300$ keV. Unfortunately, the use of POES for electron precipitation measurements have several significant problems, such
as proton contamination and varying detector efficiency that are both well-documented in the literature (e.g., Asikainen and
Mursula, 2013; Rodger et al., 2010b; Tyssøy et al., 2019). The POES data used here has been corrected for these instrumental
problems (Asikainen and Mursula, 2011; Asikainen et al., 2012; Asikainen and Mursula, 2013). In addition, the auxiliary
POES data dependent on satellite location (e.g., different coordinates, L-values, local times, and model magnetic fields) have

been reprocessed (Asikainen, 2017) to produce more accurate values than in the original POES data records. The processed
POES dataset has time resolution of 16 seconds.

This study examined the precipitating electron flux from magnetic latitudes $55°$ to $69°$ and trapped electron fluxes at L-
shells from 3.0 to 8.0. The diople model of the Earth's magnetic field is used for L-shell, and the International Geomagnetic
Reference Field is used to calculate geomagnetic latitude. Significant electron precipitation occurs at these latitudes during



magnetospheric storms due to increased electron scattering into the bounce-loss cone. However, the bounce loss cone at these

latitudes is significantly larger than the $30°$ field of view of the POES telescopes (Asikainen, 2019; Rodger et al., 2013).

This means that the POES $0°$ detectors do not resolve fluxes near the edge of the loss cone in the case of partially filled

loss cones. Additionally, the $90°$ telescope records some fluxes in the loss cone at high latitudes (Rodger et al., 2010a), so

measures both trapped and precipitating fluxes at the latitudes evaluated in this study. Therefore, we could not use the $0°$

telescope data as a direct measure of the true precipitating electron fluxes throughout the entirety of the two events. In order to

obtain a precipitating flux measurement from POES, we estimate the high-latitude precipitation as the mean of the logarithmic

fluxes recorded by the two MEPED telescopes (Eq. 1). We note that this is not a perfect measure of precipitating electron

flux ($J_{precip}$): the inclusion of the $90°$ telescope means that some trapped flux is always included in our value for $J_{precip}$,

resulting in an overestimation of precipitating flux in periods of high levels of trapped flux. This measure of $J_{precip}$ would also

underestimate precipitating flux when there are low levels of trapped flux but high levels of precipitating flux. However, this

approximation of electron precipitation from POES data has precedence in the literature (e.g., Hargreaves et al., 2010; Rodger

et al., 2013) and is useful for the qualitative analysis of electron precipitation flux undertaken in this study.

$$\log_{10}(J_{precip}) = \frac{1}{2}(\log_{10}(J_0) + \log_{10}(J_{90})) \tag{1}$$

The precipitating electron flux data is binned in 90 minute time bins and $2°$ latitude bins, and has units of counts per area

per steradian per second (cts $\text{cm}^{-2}\text{sr}^{-1}\text{s}^{-1}$). This time bin is selected to roughly correspond to the POES orbital period (102

minutes), which minimises gaps in the binned data. We use data from the three POES electron channels; $> 30\,\text{keV}$, $> 100\,\text{keV}$

and $> 300\,\text{keV}$. Note that the $> 30\,\text{keV}$ channel includes electron observations measured by the $> 100\,\text{keV}$ and $> 300\,\text{keV}$

channels, and the $> 100\,\text{keV}$ similarly contains measurements from the $> 300\,\text{keV}$ channel. In the case where no data are

recorded in a given energy channel and time / latitude bin ($J = -\infty$), the data point is set as NaN. Since multiple POES

satellites were used over a period of several days, all MLT were covered in both events. The data from multiple POES satellites

are combined by taking the mean of the measured fluxes when multiple satellites are present in the same latitude bin at a given

time.

The electron fluxes in the outer belt are analysed in four channels representing source (32.0 keV), seed (356.0 keV), core

(1.066 MeV), and ultrarelativistic (3.4 MeV) electron populations. The electron flux data is evaluated from L-shells ranging

from 3.0 to 8.0, which corresponds to minimum and maximum evaluated latitudes and encompasses the majority of the outer

radiation belt. Note that in each event the RBSP apogee is lower than $L = 8.0$ ($69°$), which constrains the available flux data.

Data at each energy level are binned in 4 hour time bins and $0.5$ L-shell bins. The RBSP orbital period is 537 minutes ($\sim 9$

hours), so a four hour time bin results in the highest time resolution with minimal gaps when using data from the two probes.

A white bin is used in the colour maps to represent a time- and L-bin where no data were recorded. The use of two RBSP

satellites over a period of multiple days meant that all MLT were encompassed, as was also the case for the POES data. Data

from multiple RBSP satellites were combined in the same way as the POES data, i.e. by taking the mean flux when both





satellites recorded fluxes in the same time and L-shell bin. There are two dates when RBSP data were not available for an instrument, as discussed above, and data from a single Van Allen Probe is used on these dates.

The binned trapped flux and precipitation flux data at different energy levels are first plotted as a function of time and location throughout the two events. The median trapped and precipitating fluxes in specific L-shell and latitude bands respectively were then investigated. The latitude ranges investigated were $55° \leq \phi < 60°$, $60° \leq \phi < 65°$, and $65° \leq \phi < 70°$, with corrected geomagnetic latitudes described above. The diople model of the Earth's magnetic field was used to calculate the corresponding L-shell ranges ($\cos\phi = \frac{1}{\sqrt{L}}$), giving ranges of $3.0 \leq L < 4.0$, $4.0 \leq L < 5.6$, and $5.6 \leq L < 8.0$. This allowed for evaluation of changes in electron trapped and precipitating fluxes during different stages of the ICME, and also allowed comparison between fluxes at a given location.

## 3  Solar Wind and Geomagnetic Conditions

### 3.1  Event 1: 30 December 2015 - 2 January 2016

The first ICME studied impacted the Earth on 31 December 2015. Figure 1 shows the conditions in the solar wind and interplanetary magnetic field (IMF), as well as geomagnetic indices, for this event over a four day period. On 31 December 2015 at 00:18 UT, the Earth is impacted by the ICME leading shock. This shock has a magnetosonic Mach number of 2.6 and a shock angle (i.e., the angle between the shock normal and upstream magnetic field) of $\theta_{Bn} = 43°$. These values are taken from the Database of interplanetary shocks, (http://www.ipshocks.fi/, Kilpua et al. (2015b)). The shock is followed by an ICME impact on 31 December 2015, 17:00 UT that persisted until 2 January 2016 11:00 UT.

There is an enhanced ($\sim 15$ nT) magnetic field during the sheath region, with fluctuating magnetic field strength and large variations in the magnetic field direction, which is a significant increase from quiet time values of $\sim 5$ nT. The $Z$-component of the IMF ($B_Z$) fluctuates from north to south relatively rapidly throughout most of the sheath period. Shortly prior to the arrival time of the leading edge of ICME ejecta, the $B_Z$ exhibits a strong, prolonged ($\sim 4$ hours) southward excursion that reaches $\sim -15$ nT. At the leading edge of the ICME ejecta, the magnetic field magnitude is $\sim 15$ nT and this smoothly decreases to $\sim 5$ nT by the end of the ejecta period. There is also a smooth, coherent magnetic field rotation from south to north during the ejecta period. The ejecta magnetic field is initially strongly southward ($B_z \approx -10$ nT) and rotates throughout the first half of the ejecta period to become northward ($B_z \approx 5$ nT) in the latter portion of the ejecta. $B_z$ then decreases to fluctuate around $\sim 0$ nT at the end of the ICME. This cloud thus shows a South-North (SN) rotating magnetic field.

The solar wind speed is low ($v_{sw} < 400 \mathrm{kms}^{-1}$) prior to the shock impact. There is a rapid increase in solar wind speed during the sheath region, which peaks at $v_{sw} \approx 500 \mathrm{kms}^{-1}$ several hours after the shock impact. The solar wind speed remains relative low, $v_{sw} <\sim 500 \mathrm{kms}^{-1}$, for the rest of the investigated period. Solar wind dynamic pressure is low before the ICME impact ($\sim 2$ nPa) and the magnetopause is reasonably stable at $\sim 10 R_E$. At the shock, dynamic pressure increases and varies between $5 - 10$ nPa in the sheath. The magnetopause rapidly shifts to $\sim 8 R_E$ at the shock impact, and then moves Earthward during the sheath region period, finally reaching geostationary orbit as a response to the compression by higher solar wind dynamic pressure and erosion due to southward IMF (Shue et al., 1998). Upon ejecta impact, there is an abrupt decrease in dynamic

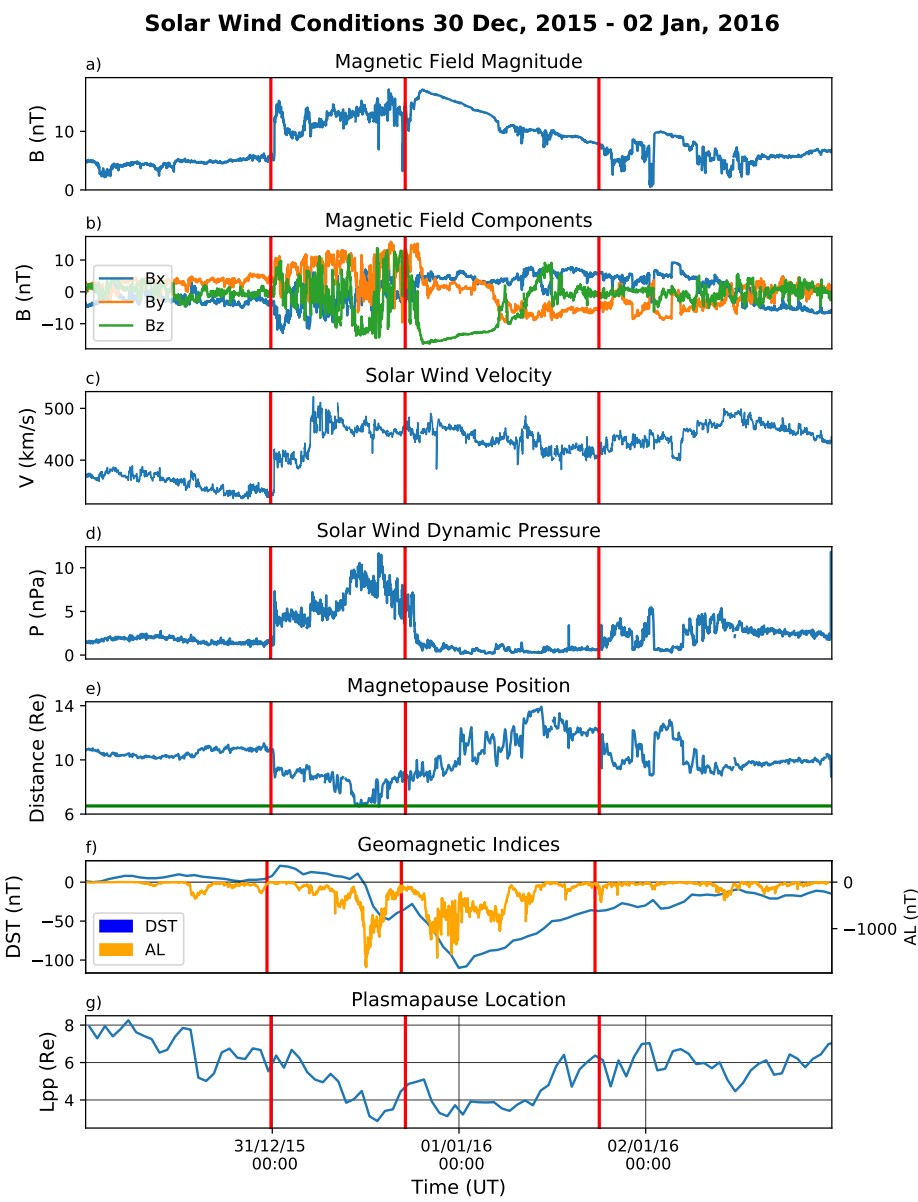

**Figure 1.** Solar wind and interplanetary magnetic field conditions during the 30 Dec., 2015 - 2 Jan., 2016 event. The red lines indicate the times when the shock, ICME ejecta impact, and ICME end occurred. These times divide the data into a pre-event period, sheath region, ICME proper, and post-event period. A clear South-North magnetic cloud rotation can be observed in subplot b. The green line subplot e shows the location of geostationary orbit.

225    pressure ($\sim 1$ nPa) and the magnetopause moves further away from Earth. The magnetopause expands to almost $14R_E$, which is considerably further from Earth than its nominal location, and this large expansion is due to low dynamic pressure combined

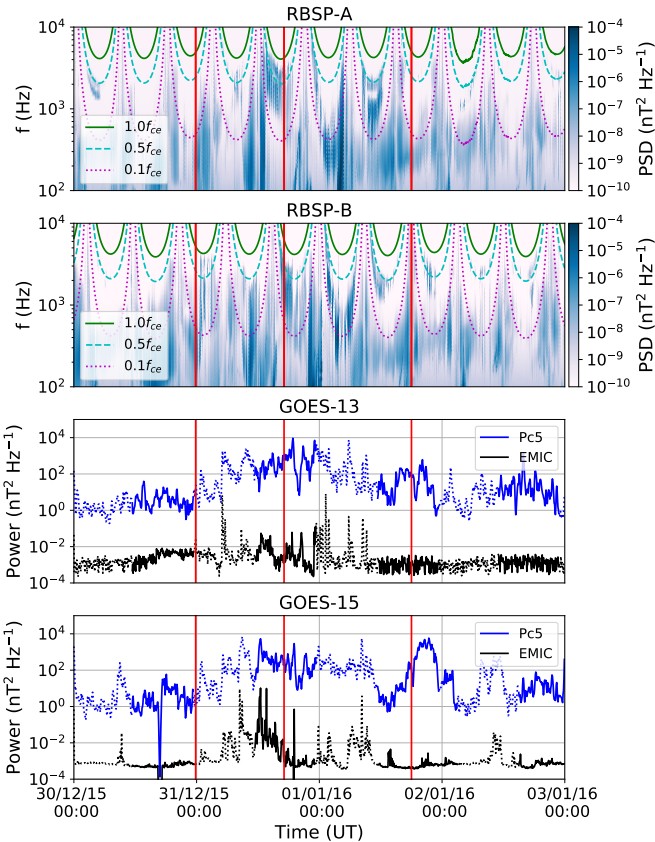

**Figure 2.** Wave power during Event 1. The upper two subplots show spectral intensity spectrograms measured by RBSP-A and RBSP-B. The region between the pink and blue dashed lines represents lower-band chorus waves, while upper-band chorus occurs between the blue and green lines. The lower two subplots show ULF Pc5 and EMIC wave powers as calculated from GOES magnetometer data. The vertical red lines again indicate the times of shock impact, ejecta impact start, and ejecta end, as shown in Figure 1.

with northward IMF in the trailing portion of the ejecta period. In the post event period, solar wind dynamic pressure enhances to 3-5 nPa. The magnetopause location fluctuates after the end of the ICME and returns to its nominal location ($\sim 10 R_E$) approximately 12 hours after the end of the ejecta.

230      The plasmapause is initially relatively far from the Earth, between $L \sim 6 - 8 R_E$. During the sheath period, the plasmapause shrinks steadily Earthward to $L \sim 2.5 R_E$. There is a short-lived expansion after the ejecta impact, but the plasmapause quickly returns to $L \sim 3 R_E$ during the southward leading fields of the magnetic cloud when there is strong global magnetospheric convection. The plasmapause remains at $L \sim 3 R_E$ for the rest of the southward potion of the ejecta period. Therefore, the majority of the outer belt is outside the plasmasphere for the majority of the time period evaluated in this study. When the field

235     turns northward and convection weakens, the plasmapause moves to $L \sim 6 - 7 R_E$ and stays at this L-range for the rest of the





investigated period. Therefore, the outer radiation belt is largely inside the plasmasphere at the end of the cloud and during the post-event phase.

Both Dst and AL indices show geomagnetically very quiet conditions before the ICME. At the shock and during the first half of the sheath region, Dst attains slightly positive values due to high dynamic pressure combined with dominantly northward IMF. Near the end of the sheath region, at the time of the extended southward fields and strong dynamic pressure, geomagnetic activity increases considerably. Dst sharply decreases to $\sim -50$ nT and AL indicates a strong substorm with a spike to approximately $-1800$ nT. There is a slight increase in Dst index and weakening of AL activity just before the ICME impact, but activity soon enhances again during the strongly southward IMF in the leading edge of the cloud. Dst drops and reaches its lowest value of $\sim -110$ nT at approximately 01:00 UT on January 1, 2016. This event thus causes an intense geomagnetic storm (i.e., Dst less than -100 nT). The AL index again shows strong high-latitude activity during the ejecta period as it fluctuates between $\sim -300$ and $-1300$ nT. The storm recovery phase begins when the field turns northward, as shown by the AL and Dst recovery. Dst increases steadily for the remainder of the ICME ejecta period and returns to quiet time values by the end of the analysed time period. AL recovers faster than Dst, and there is very minimal substorm activity during the post-ICME period.

Next, we describe the wave activity observed in the inner magnetosphere, as shown in Fig. 2. There are initially low levels of chorus, hiss, ULF Pc5, and EMIC activity prior to the sheath impact. Chorus and hiss waves are enhanced during the sheath region, with particularly strong enhancements during the period of extended southward IMF. The ULF Pc5 and EMIC wave power also increase during the sheath region period and remain enhanced during the ejecta period. Lower-band chorus and hiss waves are strong during the southward portion of the ejecta phase (see RBSP-B in particular) and there is also some upper band chorus wave activity observed at this time. The enhanced lower-band chorus and hiss waves persist into the time period of the northern portion of the ejecta. ULF Pc5 wave power weakens during the time period of the northward portion of the ejecta, but stays at relatively enhanced levels that persist for a few hours after the ICME ended, while EMIC power decreases to pre-event levels. After the end of the ICME, chorus waves are not observed but weaker hiss continues throughout the rest of the evaluated time period, consistent with RBSP being mostly inside the plasmasphere.

## 3.2 Event 2: June 26 - 30, 2013

Figure 3 shows the conditions in the solar wind, interplanetary magnetic field and geomagnetic field during Event 2. This ICME also has a rotating magnetic field, but now the field rotation is from North to South (in contrast to the South-North rotation of Event 1). The shock driven by this ICME impacts the Earth on June 27, 2013 at 13:51 UT. It is a quasi-perpendicular shock with $\theta_{Bn} = 74°$ and magnetosonic Mach number 2.0, i.e. considerably weaker than the shock in Event 1. The ICME ejecta occurs from June 27, 2013, 02:00 UT to June 29, 2013, 12:00 UT

Prior to the shock, IMF magnitude is low ($B \approx 3$ nT), but the magnetic field strength increases abruptly to $\sim 7$ nT at the shock impact. The IMF magnitude increases gradually throughout the sheath but remains below 10 nT with strongly fluctuating magnetic field direction. The magnetic field magnitude remains enhanced and relatively constant ($\sim 11$nT) throughout the ICME ejecta. There is strongly northward ($B_z \sim +10$nT) magnetic field at the leading edge of the ICME, which rotates

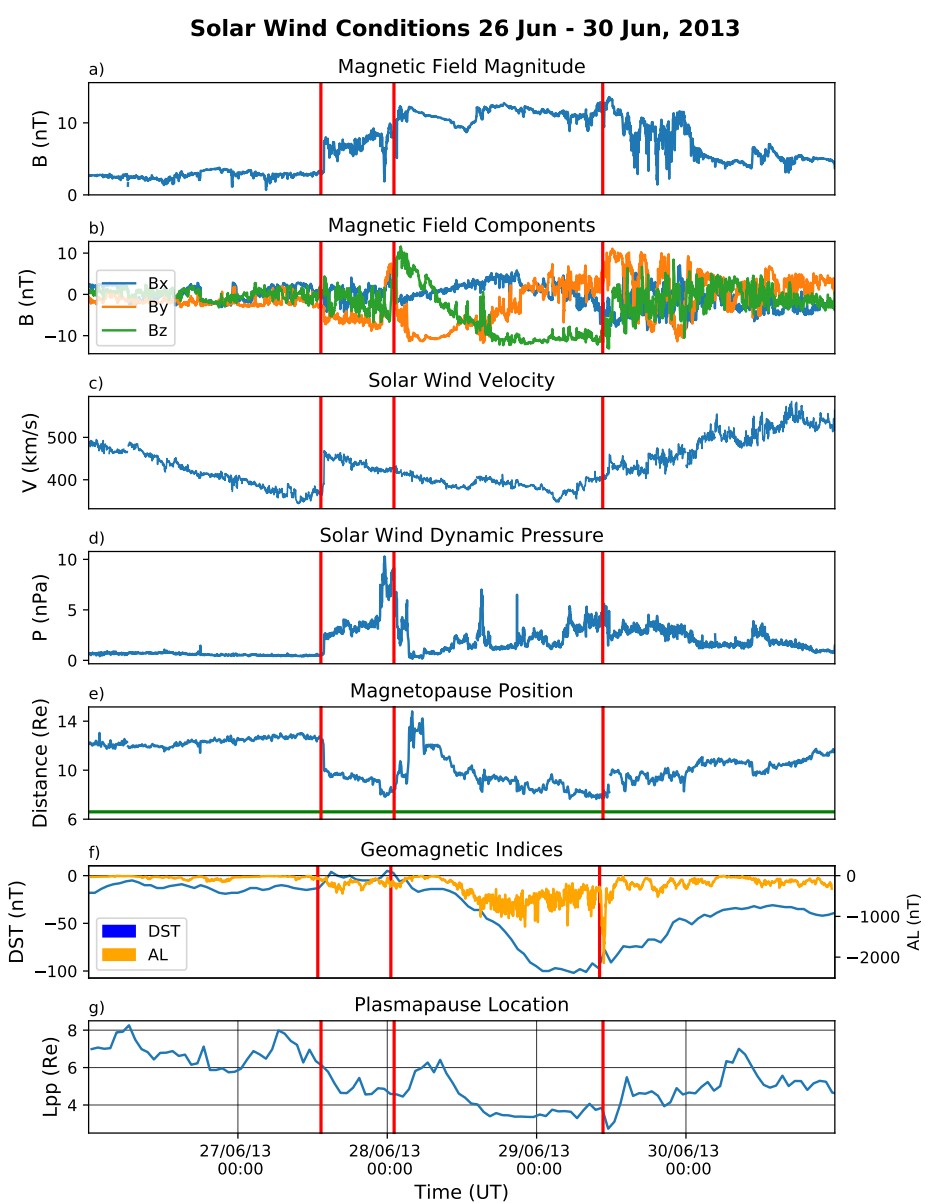

**Figure 3.** Same as Fig. 1 but for Event 2. Subplot b now shows magnetic cloud rotation from North to South.

southward to reach ∼ −10nT near the midpoint of the ICME. The field remains strongly southward during the latter half of the ICME. In the post-event period, there are large fluctuations in the magnetic field strength and direction for ∼12 hours before the IMF stabilised at ∼ 5 nT, which is slightly elevated when compared to the pre-event IMF strength. The total magnetic field strength and $Bz$ magnitude in the Event 2 ejecta reaches similar maximum values as in Event 1.

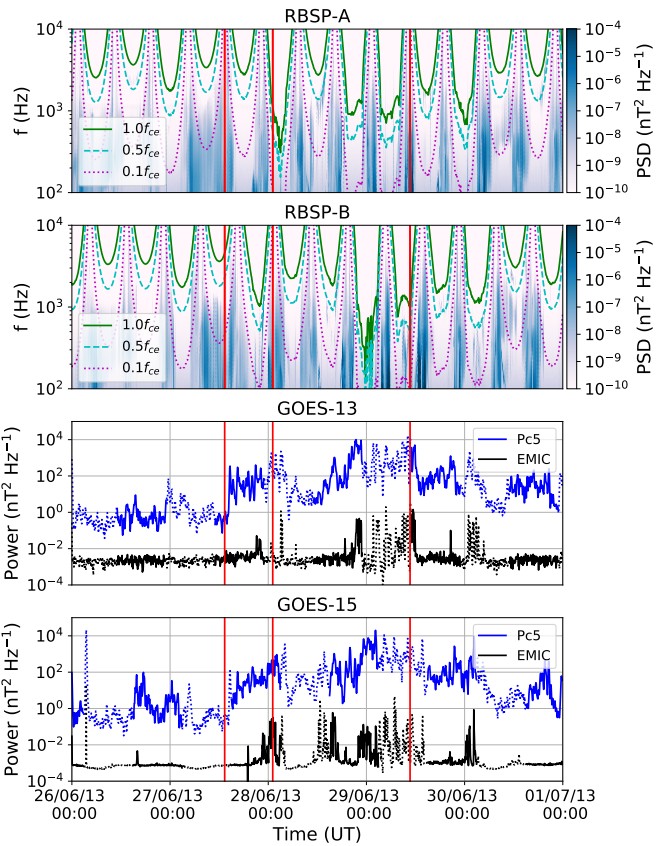

**Figure 4.** Same as Fig. 2 but for Event 2.

The solar wind speed shows a decreasing trend in the pre-event phase, declining from $\sim 500 \, \mathrm{kms}^{-1}$ at the start of the

investigated period to $\sim 350 \, \mathrm{kms}^{-1}$ immediately before the shock impact. The shock impact causes an abrupt increase in solar wind speed to $\sim 490 \, \mathrm{kms}^{-1}$, which is then followed by a steady decrease throughout the sheath and ejecta period. The solar wind speed reaches $\sim 350 \, \mathrm{kms}^{-1}$ near the end of the ejecta period and then increases throughout the post-event period to reach $\sim 600 \, \mathrm{kms}^{-1}$. The solar wind dynamic pressure is low ($< 1$ nPa) in the solar wind preceding the ICME and the magnetopause is far from the Earth ($\sim 13 R_E$). The dynamic pressure increases at the shock, and peaks at $\sim 10$ nPa near the

end of the sheath region. The magnetopause is compressed from $\sim 13 R_E$ to $\sim 10 R_E$ at the shock, and then further compresses throughout the time period of the sheath region until it reaches $\sim 8 R_E$ immediately prior to the time of ejecta impact. Maximum magnetospheric compression occurs at the time of peak solar wind pressure. During the cloud, there is again considerably lower dynamic pressure than during the sheath region, with several spikes up to $\sim 5$ nPa. This, along with northward magnetic fields in the leading part of the ejecta, results in rapid outward relaxation of the magnetopause to $\sim 14 R_E$. However, the magnetopause

is compressed increasingly closer to the Earth during the ejecta period due to increasingly strong southward magnetic fields, finally reaching $\sim 8 R_E$ by the end of the ejecta time period. The dynamic pressure remains elevated in the post-event period





when compared to pre-event values, but is generally low. The magnetopause position slowly increases in the post-event period, but does not return to its nominal position by the end of the analysed time period.

The plasmapause is initially located at $L \approx 6 - 8R_E$ during the quiet period preceding Event 2, which is similar pre-event

location as observed in Event 1. Like in Event 1, the plasmapause is compressed throughout the sheath region. However, the plasmapause remains at $L > 4R_E$ during the sheath region of Event 2, which is more distant from the Earth than seen in Event 1. The plasmapause relaxes during the time period of the northern portion of the ejecta, reaching $L \sim 6R_E$, before being compressed to $L < 4R_E$ during the latter, southern half of the ejecta. Plasmapause recovery is slower in Event 2 than Event 1, likely due to the southward magnetic fields persisting after the trailing edge of the ejecta, and the plasmapause does not return

to nominal location within the evaluated time period.

Event 2 is also associated with an intense geomagnetic storm. Geomagnetic activity is quiet before the shock impact, in terms of both Dst and AL, and remains low throughout the sheath region. Dst begins to decrease rather steadily soon after the ejecta leading edge arrives at Earth, reaching a minimum value of $-102$ nT on June 29, 2013, 07:00 UT, a few hours after the mid-point of the ejecta. This geomagnetic storm is of the same magnitude as the storm caused by Event 1. AL also decreases

during the latter, southward half of the ICME period, fluctuating between $-500$ nT and $-1000$ nT. There is a large negative spike in AL coinciding with the time of the ICME trailing edge to $\sim -2000$ nT. The recovery phase of the storm starts at the arrival time of the ejecta trailing edge and is prolonged due to large IMF fluctuations in the post-ICME period. The AL activity subsides considerably in the recovery phase, showing only weak activity with peak values around $-200$ nT in the post-event period. Dst activity remains elevated throughout the post-event period.

Figure 4 shows that there are initially very low levels of chorus, hiss, ULF Pc5, and EMIC wave activity in Event 2, which is also the case in Event 1. The chorus and hiss wave activity remains strongly suppressed throughout the sheath region and northern portion of the ejecta, except for the occurrence of some plasmaspheric hiss. The ULF Pc5 power is enhanced during the sheath period, while EMIC activity is enhanced for a short period at the sheath-ejecta boundary. The chorus and hiss wave power remains very low during the ejecta, even during the times of southern IMF, which is markedly different to the

strong enhancements that are observed during Event 1. By contrast, the strongest ULF Pc5 and EMIC power occurs during the southward IMF period in the latter half of the ejecta period. After the ejecta period, the chorus wave activity remains very low and there are periodic enhancements of plasmaspheric hiss activity. EMIC wave activity declines after the ICME, showing only some a very sporadic enhancement, while ULF Pc5 wave intensity remains elevated.

## 4    Electron Flux and Precipitation

We will now investigate the changes in electron flux precipitating into the upper atmosphere and the variations in the electron fluxes trapped in the outer Van Allen radiation belt. The variations in trapped and precipitating fluxes during Event 1 are shown in Figures 5 and 6, while Figures 7 and 8 show the electron flux dynamics during Event 2. The colour plots (Figures 5 and 7) show the variations of the precipitating electron flux (upper three subplots) and trapped electron flux (lower four subplots) over





time and space during the ICMEs. The median precipitating and trapped fluxes for three selected latitude bands are shown in

Figures 6 and 8. Section 2 details the methods and energies / latitudes / L-shell ranges used in this analysis.

### 4.1    Event 1: 30 December 2015 - 2 January 2016

Figure 5 shows the precipitating electron fluxes determined from POES measurements (Eq. 1) and the trapped electron fluxes in the outer belt measured by the Van Allen Probes. The evaluated time period is from 30 December 2015, 0 UT to 2 January 2016, 24 UT, which encompasses the day prior to the leading shock impact to the day after the impact of the trailing edge of

the ejecta of Event 1.

In the pre-ICME phase, there is a moderate, relatively constant level of precipitating flux seen in Figures 6 and 7. The precipitation encompasses the majority of the evaluated latitudes, but most precipitation occurs at high latitudes ($> \sim 60°$). The median precipitation at selected latitude curves in Fig. 6 indeed show that the lowest levels of precipitating flux occur at the lowest latitude band (green dots) and also shows relatively little variation at all energies and latitude bands, although there is

some low-latitude variation in the 100 keV and 300 keV populations. The outer radiation belt trapped electron fluxes are also relatively stable prior to the shock impact, seen best in Fig. 5. The source (32.0 keV) electron flux is initially relatively low, particularly at $L \approx 4.0 - 6.0$, and the seed electrons (354 keV) have greater flux at $L > 4.5$ than at $L < 4.5$. The core (1.066 MeV) and ultrarelativistic (3.4 MeV) electron fluxes peak at slightly lower L-shells, at $3.0 \le L \le 4.0$ and $3.5 \le L \le 4.0$, respectively.

The shock impact does not immediately significantly alter precipitating or trapped electron fluxes at any of the evaluated energies. This is clearly visible in both the colour density plot (Fig. 5) and the median precipitation / flux curves (Fig. 6), which show low variation in electron dynamics for the first half of the time period of sheath region. However, there is considerable variability in electron dynamics during second half of the sheath region, with significant precipitations enhancements and depletions beginning during the mid-sheath when there is strongest magnetospheric compression and extended southward

IMF. The $> 30$ and $> 100$ keV precipitating electron populations are enhanced at high ($> 63°$) latitudes at the mid-sheath time, although the lower latitude precipitation is not significantly changed in either population. The high latitude enhancement generally intensifies and moves to lower latitudes towards the arrival time of the ejecta leading edge, extending to ($\sim 59°$) immediately prior to shock impact. The enhancement is relatively larger in magnitude for the $> 30$ keV channel than the $> 100$ keV channel. However, there is also a decrease in $> 30$ and $> 100$ keV precipitation at highest latitudes immediately before the

arrival time of the ejecta leading edge, which is particularly clear in the colour map of the $>100$ keV population. Also during the mid-sheath time, the $> 300$ keV precipitation channel is heavily suppressed, with a large, distinct decrease in precipitation that starts at the highest latitudes ($> \sim 67°$). This high-energy precipitation depletion spreads to lower latitudes ($\sim 63°$) and deepens around the time of the time of the transition from the sheath to the ejecta. However, there remains some constant $> 300$ keV precipitating flux at at lowest evaluated latitudes throughout the sheath region.

Consistent with the precipitation dynamics described above, trapped electron fluxes are relatively stable during the time of the leading part of the sheath and distinctly change during the trailing section. The changes in flux again coincide with the time period of extended southward IMF and magnetopause compression to geostationary orbit. Source electron fluxes show strong




**Figure 5.** Variation in the precipitating electron fluxes lost into the atmosphere (upper three subplots) and the trapped electron flux in the outer radiation belt (lower four subplots) observed during Event 1. This plot shows the spatial and temporal variations in those fluxes for different electron populations. Fluxes are measured in $\text{cts cm}^{-2}\text{sr}^{-1}\text{s}^{-1}$. The red lines show the times of the shock, ICME start, and ICME end, as in Figure 1. The y-axis on the right of the flux subplots shows the corresponding geomagnetic latitude, as defined by $\cos\phi = L^{-\frac{1}{2}}$. Note the different colour scales for each subplot.





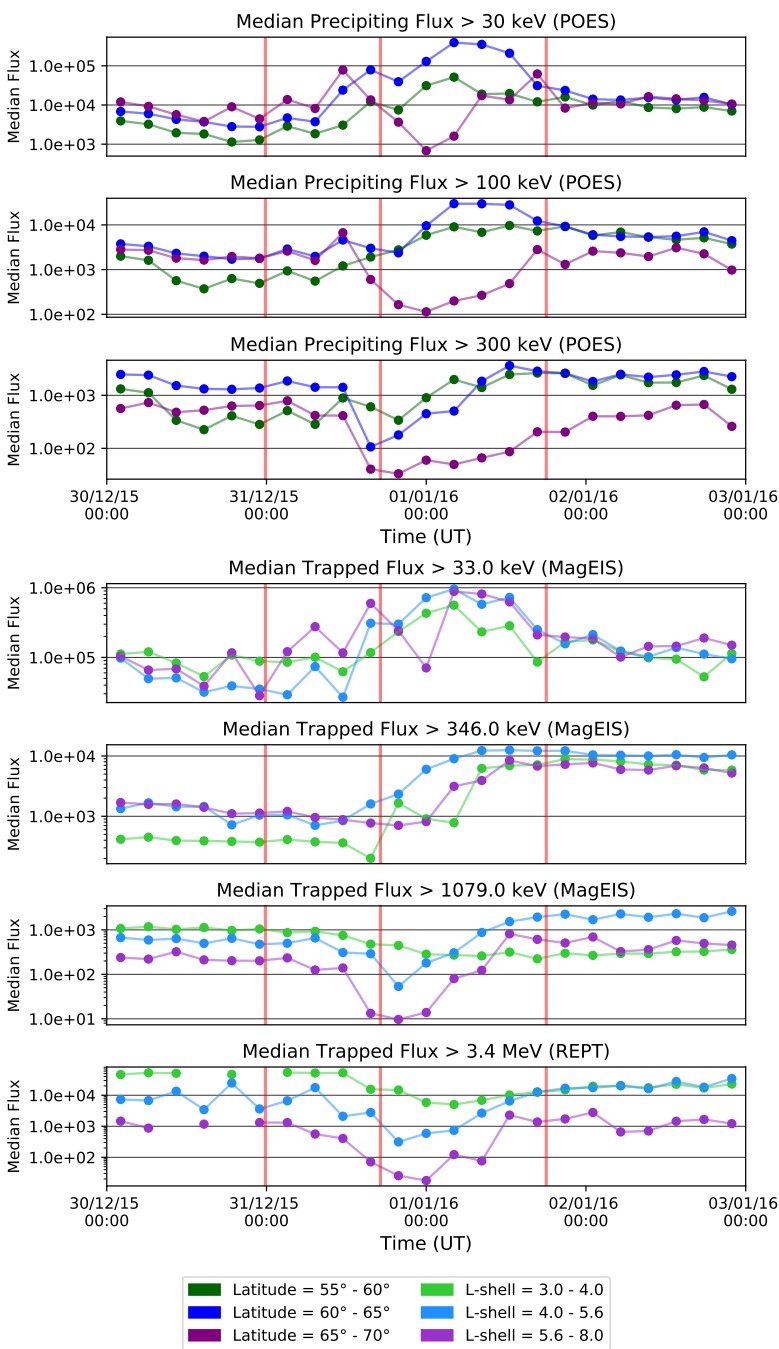

**Figure 6.** Variation in median trapped and precipitating fluxes during Event 1 at selected latitudes. Median values were calculated over four hour time intervals, and median trapped flux was calculated at L-shell corresponding to the latitude ranges used for precipitation analysis. Fluxes are measured in cts cm$^{-2}$sr$^{-1}$s$^{-1}$.





enhancements during the time period of the latter portion of the sheath region, particularly at the higher L-shells (latitudes). The seed population in turn remains more stable throughout the sheath time period with some increase in fluxes near the time of the

ejecta impact, while core and ultrarelativistic trapped electron fluxes decrease at high L-shell. This decrease in core electron flux is clear but small, with the exception of a deep decrease at the highest L-shell bin ($5.5 < L < 6.0$, see in particular Fig. 5). The ultrarelativistic electron population shows a similar deep decrease at $5.5 < L < 6.0$, and the decrease at lower L-shells is more pronounced than in the core population. These depletions extend throughout the time period of the leading portion of the ejecta, although the flux soon recovers at lower L-shell. The depletion in the ultrarelativistic populations extends as low

as $L \sim 4.0$ and persists for longer times than that seen for the core population, which is only briefly depleted to a minimum L-shell of $\sim 4.5$. The high-latitude core and ultrarelativistic trapped flux depletions thus coincide with the observed decrease in high-latitude precipitation fluxes in the $> 300$ keV channel. We note that the precipitating flux data shows the depletion extending to higher latitudes than recorded by the RBSP probes, indicating that the trapped flux depletions may extend to L-shells (latitudes) beyond the RBSP apogee (discussed in Section 2).

The strongest latitude/L-shell variations in precipitating and trapped fluxes occur during the ejecta period, as seen in both the colour density plot and median line plot. Firstly, all POES channels exhibit strong decreases in precipitating flux at the highest latitudes throughout most of the ejecta time period (purple dots in Fig. 6). As discussed previously, this depletion begins a few hours before the arrival time of the ejecta leading edge, and is particularly deep and wide in the $> 300$ keV channel. The decreased precipitation spreads somewhat in latitude as the ejecta progresses, as shown by Fig. 5. In contrast, mid-latitude

precipitation enhances for all channels, with greatest spread of enhanced precipitation at the lowest energies ($\sim 57° - 65°$ in the $> 30$ keV channel compared to $\sim 57° - 61°$ for the $> 300$ keV population). The precipitating flux magnitude is greatest at the beginning of the ejecta period for the $> 30$ keV channel, while the higher energy precipitating fluxes enhance as the ejecta progresses. The precipitating flux in the $> 30$ keV and $> 100$ keV channels slightly decreases just prior to the arrival time of the ejecta trailing edge, which is when the $> 300$ keV channel precipitation peaks. Overall, each energy channel shows an initial

high-latitude depletion that recovers throughout the ejecta time period, and additionally has a strong, coherent mid-latitude enhancement with energy-dependent timing and duration.

The outer belt electron trapped fluxes also show strong variability and similar overall trends as the changing precipitating fluxes observed during the ejecta period. For the source and seed populations, trapped flux enhancements begin slightly prior to the time of ejecta impact, as discussed earlier. Both fluxes enhance greatly over wide L-shell range ($3.0 < L < 5.5$) during

the times of the leading part of the ejecta. Only the highest L-shell bin shows a mild decrease in source and seed trapped fluxes. The source population flux is strongly decreases towards the time of the ejecta trailing edge, but the seed electron flux remains elevated in the latter part of the ejecta period. The core and ultrarelativistic electrons progressively enhance during latter portion of the ejecta period, following the seed population enhancement. The timing and magnitude of these enhancements are energy dependent, with lower energy populations being enhanced earlier and to a greater extent than higher energy populations.

The peak enhancements for the seed, core, and ultrarelativistic populations occurs at $4.0 < L < 5.5$ ($60° - 63°$), and these enhancements persist until the end of the evaluated time period. The core and ultrarelativistic fluxes are also strongly depleted during the leading portion of the ejecta period at high L-shell, as discussed above. This low latitude depletion is most distinct





in the ultrarelativistic population. The concurrent enhancements in precipitation at corresponding latitudes indicate that the precipitating populations originated from the radiation belts.

After the end of the ICME time period, the electron precipitating fluxes stabilises quickly at all energies, returning to a relatively constant level of precipitation across all latitudes. The magnitude of precipitation in all channels is, however, slightly higher when compared to the pre-event period, especially at middle latitudes. This is consistent with higher energy electron fluxes remaining clearly enhanced, which reflects a strong trapped electron population in the belts following the ICME impact and subsequently allowing for greater precipitation. In contrast, the source electron fluxes show a drastic decrease from the

values during the ejecta, although the source fluxes are still slightly greater than before the ICME. The source fluxes remain relatively stable following the ejecta period, likely due to the lack of substorm activity.

### 4.2   Event 2: 26 - 30 June, 2013

Figure 7 shows the variation in precipitation and trapped electron flux from 26 June 2013, 0 UT to 30 June 2013, 24 UT, encompassing the time period that Event 2 occurred. This event was an ICME with a North-South magnetic field rotation in

the ejecta. Prior to the ICME impact, Event 2 exhibits moderate level of electron precipitation at all evaluated latitudes and energies, similar to Event 1. We again observe that the majority of this quiet-time precipitation occurs at $\sim 63°$ (mid-latitude band, blue dots in Fig. 8). The pre-event electron fluxes also partly show similar L-shell / latitude distribution as in Event 1. We suggest that this is reasonable given that both are "quiet" or "pre-event forcing" conditions. The trapped source electron flux is very low at $4.5 < L < 5.0$ and this region of low flux expands to higher L-shells over time prior to shock impact. The seed,

core, and ultrarelativistic populations all have significant flux at L-shells $> 3.5 - 4$, and these fluxes remain relatively stable prior to the shock impact. The peak fluxes in the core and ultrarelativistic populations are now at $4.5 < L < 6.0$.

    Neither the precipitating nor the trapped electron fluxes experience major changes due to the shock impact in this event. There are also minimal changes during the period of sheath interaction with the inner magnetosphere, likely due to the lack of sustained southward IMF and minimal geomagnetic activity. Some notable changes occur, however, close to the time period of

the sheath - ejecta boundary, which is the period that coincides with the large solar wind dynamic pressure spike and magnetopause compression to $\sim 7.5 R_E$. The seed, core, and ultrarelativistic electron populations exhibit a decreasing trend at middle and high latitudes at this time, most visible in the median plot (blue and purple dots, Fig. 8), with greatest depletions taking place in highest energy populations. There are minimal precipitation changes upon shock impact, with a small enhancement in the POES $> 30$ keV channel and slight depletion in the $>100$ keV and $>300$ keV channels at the highest latitudes.

There are only small changes in electron precipitation immediately following the ejecta impact, likely due to the northward IMF at the leading edge. The $>30$ keV enhancement persists but remains at a low magnitude and is restricted to highest latitudes, and this is also the case for the depletions in the higher energy populations. During the time covering the trailing part of the ejecta, as the IMF rotates southward and geomagnetic activity intensifies, there is however a much stronger variability in electron precipitation. The $> 30$ keV and $> 100$ keV POES channels show alternating precipitation increases and decreases,

but the general trend is a significant enhancement during the ejecta edge, in particular at mid-latitudes ($60° - 65°$, blue dots in Fig. 8). The color density plot in 7 further shows that this region of enhanced precipitation widens towards lower latitudes



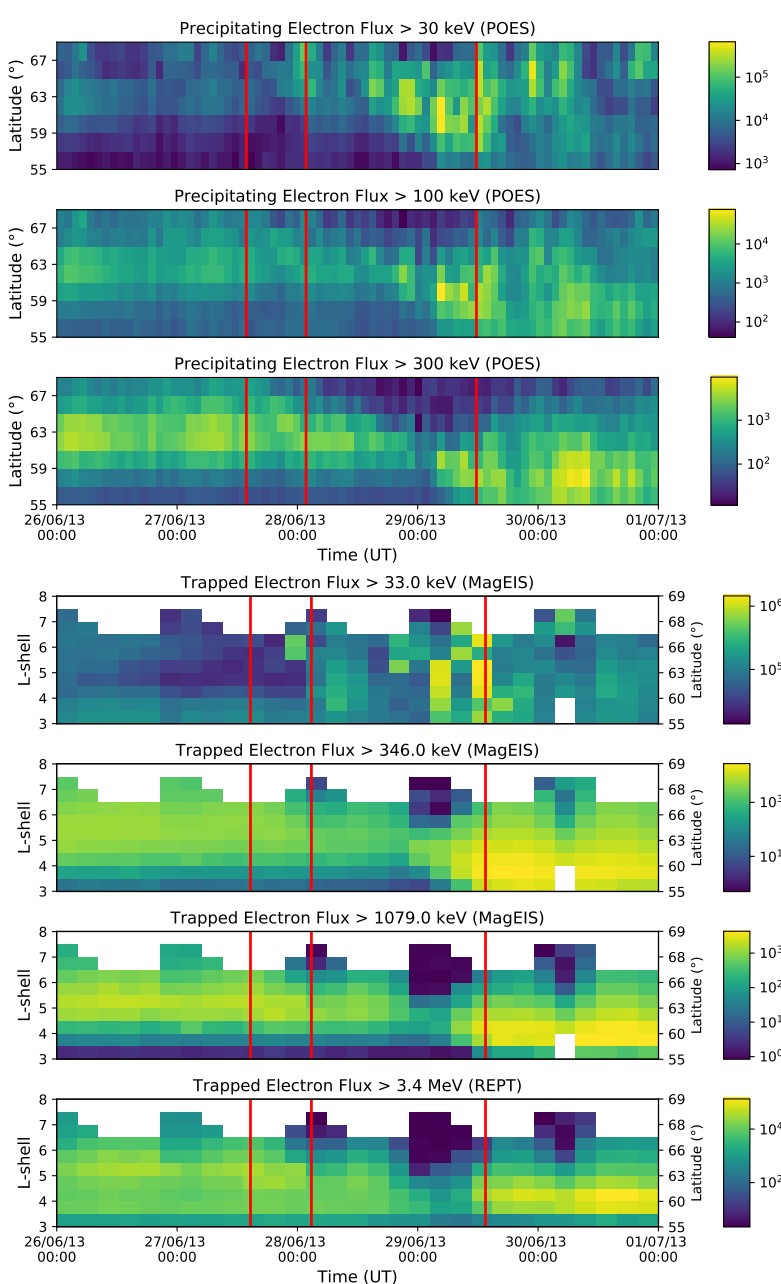

**Figure 7.** Same as Fig. 5 but for Event 2. Fluxes are measured in cts $\mathrm{cm}^{-2}\mathrm{sr}^{-1}\mathrm{s}^{-1}$.



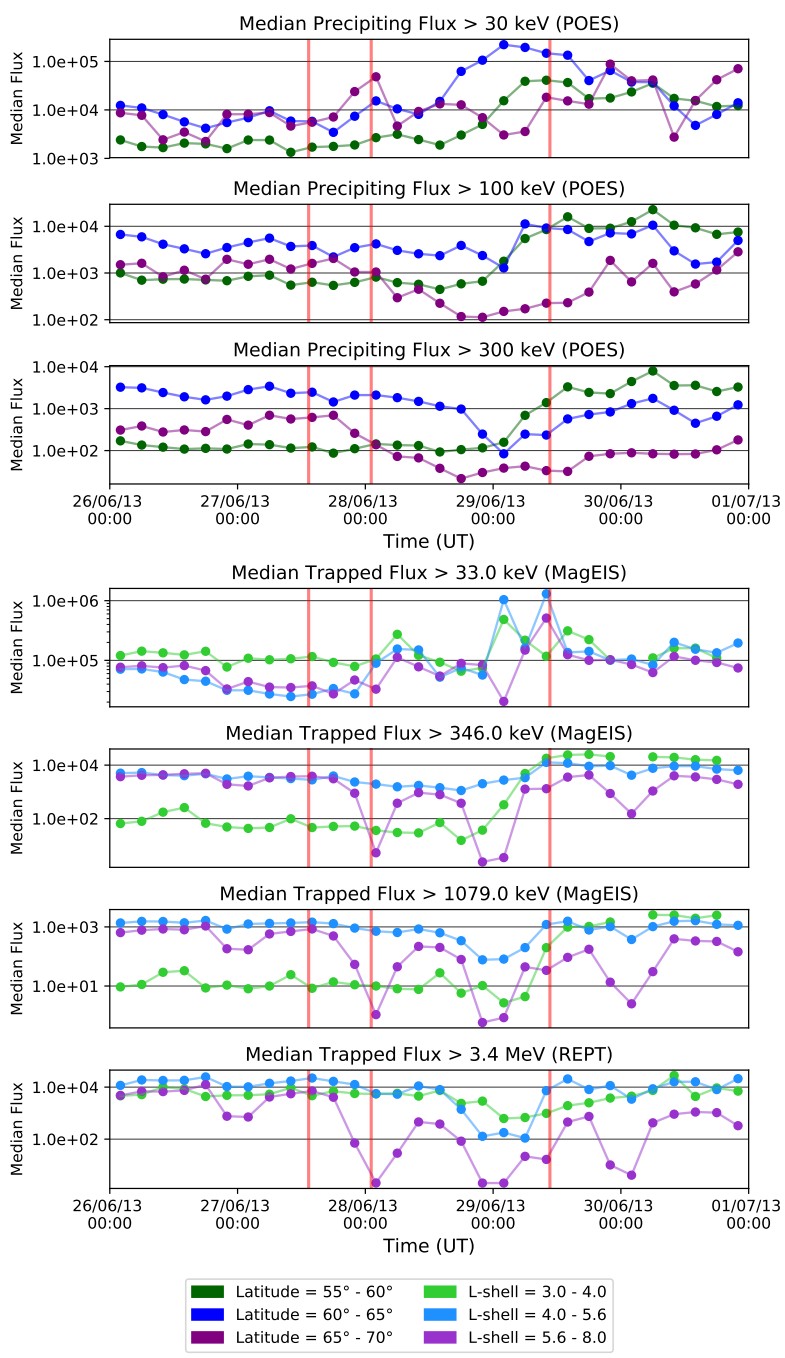

**Figure 8.** Same as Fig. 6 but for Event 2. Fluxes are measured in cts $cm^{-2}sr^{-1}s^{-1}$.





over time. These channels also shows a clear decrease in precipitation flux at the highest latitudes at the same time as the mid-latitude flux enhancement, with the >100 keV depletion extending over a wider latitude range than the >30 keV depletion. The $> 300$ keV channel shows a depletion in precipitating electrons for both the highest and mid latitudes at this time, with the

region of low precipitation fluxes extending to lower latitudes over time. At the lowest evaluated latitudes ($55 - 60°$), there is a clear precipitation enhancement towards the end of the ejecta time in the $> 300$ keV channel, seen clearly in the green dots of Fig. 8.

  The outer belt trapped electron fluxes also show the most variations during the time period covered by the trailing part of the ejecta. The source trapped electron fluxes at $L > 3$, which are very low during the sheath period as discussed above,

clearly enhance at the ejecta onset. The source fluxes experience some temporal and spatial variations during the northern portion of the ejecta period and then intensify more strongly over a wide range of latitudes before the time of the ejecta trailing edge. In contrast to Event 1, this source flux enhancement during the southward portion of the ejecta is not coherent and exhibits large temporal and spatial fluctuations. There is a also source flux depletion that is localised to the highest L-shells. Conversely, all other outer belt electron populations show a constant or slightly depleted flux in the time covering the leading

part of the ejecta. At the mid-point of the ejecta, the seed, core, and ultrarelativistic fluxes decrease slightly in the mid L-shell/latitude band, while a a strong depletion occurs at the highest latitudes. Populations with the greatest energy are depleted most strongly, and this depletion also extends over the widest L-shell range in highest energy populations. These depletions occur at the time of southward IMF and strong magnetopause compression. Immediately after these depletions, the seed, core, and ultrarelativistic fluxes begin to progressively enhance at lower L-shells ($L = 3.5 - 4.5$). These enhancements are once

again energy dependent, with the seed fluxes enhancing earlier and to a greater magnitude than the core or ultrarelativistic populations. These enhancements are consistent with the precipitation signatures during this time, indicating again that these precipitating fluxes originate from the outer radiation belt.

  The recovery phase of the storm related to Event 2 starts approximately at the arrival time of the trailing edge of the ejecta, but some geomagnetic activity continues, mostly likely due to the IMF remaining enhanced with fluctuating direction and

increasing solar wind speed. The magnitude of the precipitating and trapped fluxes are now generally significantly greater than before the shock impact and exhibit strong spatial variations. The source electron fluxes weaken after a strong enhancement at the time of the ejecta trailing boundary, although source fluxes greater than the pre-event levels. The peak seed and core trapped fluxes occur at lower latitudes/L-shells after the ICME than in the pre-event period, with peaks at $L = 3.5 - 4.5$ ($57° - 63°$). The ultrarelativistic flux is now also strongly peaked at $L = 3.5 - 4.5$ ($57° - 63°$), in contrast to its more spatially uniform

distribution prior to the event. We also note that at the highest evaluated L-shells/latitudes, the seed, core, and ultrarelativistic fluxes are significantly lower than the pre-event fluxes. The precipitation is strongest at low- and mid-latitudes in the $> 100$ keV and $> 300$ keV channels, while the $> 30$ keV population is stronger at highest latitudes after the ICME. One significant phenomena observed in the recovery period is the enhanced precipitating fluxes of $> 300$ keV electrons at $L < 5$, which occurs at the same time and location as a strong seed flux enhancement. The plamsapause is located at $L \sim 5$, so this high energy

precipitation enhancement is likely due to the seed population being scattered into the loss cone by plasmaspheric hiss.





## 5 Discussion

We have studied here the electron response to two different ICME events, examining fluxes that precipitate into the upper atmosphere and fluxes trapped in the outer radiation belt. Both events consisted of a shock and sheath followed by a magnetic cloud. In Event 1, the magnetic field rotated from south to north (SN) within the cloud, while the magnetic cloud in Event

2 had opposite polarity, i.e. the field rotation was from north to south (NS). Both events were associated with an intense magnetic storm with similar peak magnitudes (Dst of $-110$ nT and $-102$ nT, respectively). Each ICME induced considerable variations in precipitating and trapped electron fluxes, particularly during the time period encompassing the cloud. Both events also resulted in clear changes between pre-event and post-event conditions. However, we found several clear differences in the characteristics of the flux response that could be related to the different magnetic polarity between the investigated clouds.

For both events, the trapped source electron fluxes experienced approximately one order of magnitude increase during the storm main phase that was followed by a considerable weakening as the storm subsided. However, the timing of these variations with respect to the ICME were highly different. For Event 1, strong enhancements occurred from the time of the mid-sheath onwards and source electrons were present in most L-shells by the peak of the storm (i.e., soon after the arrival of the cloud's leading edge). These fluxes weakened only a few hours before the time of the cloud's trailing edge. In Event 2, the ejecta

impact resulted in only a small and short-lived increase in source fluxes and the main source enhancement occurred around the cloud's trailing boundary. Compared to Event 1, Event 2 also caused more localized and periodic enhancements of source electrons at different L-shells (in particular at mid L-shells). Therefore, the greatest source flux enhancements occurred during the period of southward fields in the magnetic clouds, with an additional smaller enhancement upon the sheath-ejecta transition in both events.

The two events had also similar overall seed electron flux response: fluxes in this energy range enhanced strongly as a response to the ICME and the enhancement lasted throughout the investigated post-event period. Some clear differences were again found in the seed flux response. In Event 1, the low fluxes in the pre-event phase began to enhance in the end of the sheath region. However, in Event 2, the initially significantly greater seed flux at high L-shells were first depleted (up to two orders of magnitude) and then progressively enhanced and moved to lower L-shells by the end of the cloud. The peak seed

flux enhancements occurred at $L \approx 4 - 4.5$, and $L \approx 3.5 - 4$ for Events 1 and 2 respectively, so seed flux in Event 2 peaked at slightly lower L-shells. Event 2 also resulted in a significantly larger seed flux enhancement at the lowest L-shells than occurred in Event 1, with significant flux as low as $L = 3$. The initial wide depletion of seed fluxes and the appearance of fluxes at lower L-shells suggests that genuine changes in the seed population occurred during both events. The persistence of seed enhancements in both events is significantly different to the behaviour of the source flux, which rapidly weakened in the

storm recovery phase. This suggests slower decay times due to plasmaspheric hiss (order of days, e.g., Meredith et al., 2006; Summers et al., 2008) and/or continuing acceleration to hundreds of keV energies. In both events, hiss waves were observed and ULF Pc5 wave power was at higher levels than in the pre-ICME phase throughout the storm recovery. For Event 2, there were also chorus waves present and geomagnetic activity subsided at a slower pace due to the perturbed region after the end of the magnetic cloud. The longer period of chorus activity in Event 2 could explain the more efficient seed flux enhancement.





Another distinct feature of Event 2 is a strong depletion of seed and source fluxes at the highest L-shells, which is visible during the sheath – ejecta transition and is more clearly in the end of the ejecta as the storm peaks. This was not observed in Event 1, but this could be due to the RBSP satellite apogees not covering the highest L-shells during this event. We discuss these depletions in more detail below.

    In both events, there were low, constant levels of precipitating flux in all POES energy channels prior to the ICME impact.
These low variations in precipitating fluxes during the pre-event phase are consistent with low geomagnetic activity, which is supported by the magnetopause being far from geostationary orbit and lack of significant magnetospheric wave activity. Each event triggered significant changes in precipitating flux during the sheath and / or ejecta of the ICME. The observed precipitating fluxes from the POES > 30 keV and > 100 keV energy channels roughly followed the variations in source and seed radiation belt populations for both events. We note, however, that the > 100 keV precipitation enhancement was much
more pronounced in Event 1 than in Event 2, and that > 100 keV precipitation enhancement preceded increases in seed fluxes for both events (and even preceded source fluxes in Event 1). The observed precipitation flux enhancement occurring prior to the trapped flux enhancement is likely not due to a true increase in precipitating flux at this time, but rather due to the 90° POES telescope measuring the trapped flux enhancement before it was measured by RBSP. Although the precipitating fluxes overall followed the trends in source/seed fluxes, they also featured some further differences between two events. For
Event 1, strong precipitating fluxes began during the mid-sheath period and quickly spread to lower latitudes, while Event 2 precipitation enhancements started during the middle of the cloud and progressed more gradually in both magnitude and latitude range. These differences can be attributed to the field very abruptly becoming strongly southward at the leading edge of the cloud in Event 1, while there was a more gradual intensification of the southward IMF component in Event 2. For both events, the precipitating fluxes reached the lowest latitudes at approximately the same time as the peak in source flux, and
weakened when the storm recovered, returning close to pre-ICME levels. The observed behaviour of precipitating fluxes and trapped source flux during the ejecta is consistent with the results of Turner et al. (2019), a large statistical study that analysed 110 storms during the RBSP era.

    During the main phase of the storms, injected source electrons gave rise to lower band chorus waves that effectively precipitated > 30 keV electrons from the belts (e.g., Lam et al., 2010). Therefore, enhanced > 30 keV precipitation arose both from
excitation of chorus waves that scattered keV electrons into the loss cone and the presence of a significant radiation belt population available for scattering. These source electrons likely originated primarily from substorm injections or global convection. Chorus waves were indeed observed for both events, particularly during Event 1. In Event 2, the plasmapause was further from Earth, so plasmaspheric hiss likely also made a significant contribution to the scattering. We also note that the close proximity of southward fields in the sheath and cloud for Event 1 were important for the event characteristics. This resulted in a long-
lasting and smoother source flux enhancement, and caused enhanced > 30 keV precipitating flux over a larger L-shell/latitude range that continued for the majority of the cloud. For Event 2, NS rotation caused more sporadic enhancements in trapped and precipitating flux that occurred only close to the cloud's trailing boundary. The continued trapped source flux enhancements after the ejecta in Event 1 resulted from mild geomagnetic activity, which is associated with trailing solar wind interacting with the southward fields in the cloud.





The core and ultrarelativistic populations showed significant changes in their peak flux locations and magnitudes during both events, and also exhibited strong depletions at the largest L-shells. The peak fluxes in Event 1 at these energies moved from low ($3.5 < L < 4.5$) to higher ($4.5 < L < 5.5$) L-shells, while the fluxes in Event 2 had exhibited opposite behaviour, shifting from about $5.0 < L < 5.5$ to $3.5 < L < 4.5$. The peak fluxes attained approximately similar magnitudes in both events, but fluxes in Event 2 experienced more variability following the end of the ejecta period, consistent with on-going chorus activity.

The strongest core and ultrarelativistic enhancements started earlier with respect to the ICME structure for Event 1, around the time of the middle of the cloud, while Event 2 fluxes enhanced at the cloud's end boundary time. There were progressive enhancements of core and ultrarelativistic fluxes in both events, but this occurred much more rapidly in Event 2 than in Event 1, where the enhancements occurred throughout the ejecta period. Event 1 behaviour is thus consistent with whistler mode chorus waves progressively accelerating seed electrons to MeV energies (Jaynes et al., 2015), and indeed, significant chorus wave

activity was present through most of the ICME period. The chorus wave energisation likely worked together with energisation by the inward transport by ULF Pc5 waves (e.g., Jaynes et al., 2018; Ma et al., 2018). This energisation occurred when the plasmapause was close to Earth and $L$-shells investigated resided outside the plasmasphere. For Event 2, the inward transport could be particularly important as there was high ULF activity and much weaker chorus waves activity throughout the storm main phase. ULF wave acceleration has indeed been invoked to cause fast energisation of electrons to relativistic energies

at wide L-ranges, including also at L-shells as low as $L = \sim 3$ (e.g., Jaynes et al., 2018; Kanekal et al., 1999; Mathie and Mann, 2001; Rae et al., 2019). The peak flux region at $L = 3.5$ was indeed just outside the plasmapause during the end of the cloud, so this is one possible explanation for the rapid core and ultrarelativistic enhancements observed at the end of the magnetic cloud. In Event 1, the magnetopause was located far from Earth in the latter portion of the magnetic cloud and then returned to its initial location after the ejecta. This magnetosphere expansion combined with the extremely low dynamic

pressure throughout the ejecta allowed the radiation belts to expand outwards, therefore enabling a shift in high energy fluxes to higher L-shell. Conversely, there was elevated dynamic pressure in the ejecta of Event 2, particularly near the trailing edge, with significant spikes of higher pressure. This meant that the Event 2 magnetopause was strongly compressed during the latter half of the ejecta and then remained compressed after the ejecta when compared to its pre-event location. This inward motion of the magnetopause is reflected by the inward motion of peak high energy fluxes in Event 2. The relationship between magnetopause

location and peak core and ultrarelativistic fluxes in the recovery stage of these two events highlights the importance of the dynamic pressure in determining the radiation belt location.

       The seed, core, and ultrarelativistic electron fluxes experienced a strong, high latitude depletion during the southern portion of the cloud for both events. For Event 1, the depletion started already during the sheath and extended to lower latitudes with increasing energy during the ejecta period. The precipitating fluxes at all energy channels showed similarly wide depletions

at the highest latitudes during the time of the ejecta. This suggests that depletions in outer belt fluxes extended to even higher L-shells than captured by the RBSP orbit (at least up to $L \sim 8$). These high-latitude precipitating flux measurements give particularly useful insight into the outer regions of the radiation belt in Event 1 when the RBSP apogee was at lower L-shell. Magnetopause shadowing is the most probable cause of this high-latitude depletion (Bortnik et al., 2006; Turner et al., 2014) as there was strong magnetopause compression/erosion during the period of southward ejecta in both events. In Event 2, the





560 short-lived depletion at high L-shells/latitudes upon ejecta impact is also presumably due to magnetopause shadowing, as strong magnetopause compression occurred at this time. Losses due to magnetopause shadowing were likely enhanced by the Dst effect, as the observed depletions coincided with the storm main phase/peak (e.g., Gokani et al., 2019; Turner et al., 2014). The energy-dependent spatial variations in flux depletions are likely due to inward radial gradients causing MeV electron loss at lower L-shell via outward radial diffusion (e.g., Bortnik et al., 2006; Turner et al., 2013). Radial gradients can be induced

565 by losses at high L-shell, which may have occurred above the RBSP apogee. Losses at lower L-shells are likely related to mechanisms other than magnetopause shadowing. These low L-shell depletions mainly occurred during the storm main phase, but also occurred at the time of sheath-ejecta boundary. They could therefore be caused by a combination of scattering into the bounce-loss cone due to wave-particle interactions (most likely due to EMIC waves, e.g., Turner et al., 2019; Yuan et al., 2013), outward radial diffusion by Pc5 waves, and Dst effect transporting electrons further out to locations where they may undergo

570 magnetopause shadowing. The EMIC and Pc5 wave powers were indeed high for both events at these times, indicating that interactions with these waves may have significantly impacted radiation belt electron dynamics.

 In each event, we observed that the $> 300$ keV precipitating fluxes roughly follow the variations in the seed, core, and ultra-relativistic populations. The location of $> 300$ keV precipitating flux corresponds to the location of peak high-energy trapped flux, implying that the MeV radiation belt populations at these locations provides a source for $> 300$ keV precipitation. The

575 timing of changes matches best with that of the core and ultrarelativistic fluxes, indicating that these populations experienced losses via precipitation during the southern portion of the magnetic cloud. Some recent studies indeed suggest that only a very small fraction of radiation belt electrons would be subject to losses into the upper atmosphere (e.g., $< 0.5\%$ of total losses of relativistic fluxes at $3.0 < L < 4.5$ on 17/3/2015 were due to precipitation, Gokani et al., 2019). If this is the case in these two events, it implies that the losses via enhanced precipitation would be outweighed by the gains in the trapped radiation belt

580 populations during the ICMEs that were discussed above.

## 6 Summary

This paper highlights the similarities and differences in the electron response to two ICMEs with magnetic clouds of opposite polarities. Both events resulted in similar overall changes in the source, seed, core, and ultrarelativistic trapped populations of the outer radiation belt, affecting the magnitude and location of these fluxes. However, significant differences in the electron

585 response occurred during the sheath and ejecta periods of the two events. This implies that different key energisation mechanisms acted during the events and that a given electron population may have experienced different enhancement mechanisms in each event. These differences included the timing of trapped / precipitating flux variations with respect to the ICME structures and the coherence of flux enhancements / depletions. The location of the southward field in the magnetic cloud is a key factor associated with these differences, and southward fields within the sheath region and disturbance region after the cloud

590 are likely also significant. The temporal profile of the southward field component likely also plays a role, as there was a sharp southward turn at the leading edge of the SN-cloud versus a smooth rotation to the southernmost field for NS-type cloud. Another significant factor in the electron response is the magnetopause location and dynamic pressure during the sheath and





ejecta of the two events. We also emphasize that the POES precipitating flux data roughly reflected the changes in the outer belt trapped electron fluxes at corresponding energies and locations, as recorded by the RBSP. These precipitation data provide

insight into the changes in trapped radiation belt fluxes, allowing for estimation of variations occurring at L-shells beyond the RBSP apogee and providing greater spatial and temporal resolution of the trapped flux variations in the outer radiation belt.

*Data availability.* All RBSP-ECT data are publicly available at the website http://www.RBSP-ect.lanl.gov/, and all RBSP-EMFISIS data are publicly available at the website https://emfisis.physics.uiowa.edu Processed POES data was obtained through the PROSPECT project (321440, Academy of Finland, Timo Asikainen). GOES data was obtained from the website https://www.ngdc.noaa.gov/30stp/satellite/

goes/index.html. Wind and OMNI data were obtained from CDAWeb, https://cdaweb.gsfc.nasa.gov/index.html/. Dst data are publically available from the website http://wdc.kugi.kyoto-u.ac.jp/. ICME times were obtained from the Cane and Richardson online database http://www.srl.caltech.edu/ACE/ASC/DATA/level3/icmetable2.htm.

*Author contributions.* HG carried out the data analysis, prepared the plots, and interpreted the results under the supervision of EK and AO. TA provided processed POES data. TA and CR assisted in analysis and interpretation of the POES data. MK carried out the wave analysis.

MP participated in interpreting the results and commented the manuscript. HG prepared the manuscript with contributions from all authors.

*Competing interests.* The authors declare that they have no conflicts of interest.

*Acknowledgements.* The results presented here have been achieved under the framework of the Finnish Centre of Excellence in Research of Sustainable Space (FORESAIL; Academy of Finland grant numbers 312390 and 312351), which we gratefully acknowledge. The work of TA is supported by the Academy of Finland grant 321440. The work of MP is supported by the European Research Council Consol-

idator grant 682068-PRESTISSIMO and the Academy of Finland grant 309937. The work of EK has received funding from the European Research Council (ERC) under the European Union's Horizon 2020 research and innovation programme (ERC-COG 724391). EK Kilpua acknowledges Academy of Finland project SMASH no. 310445. Processing and analysis of the ECT, MagEIS and REPT data was supported by Energetic Particle, Composition, and Thermal Plasma (RBSP-ECT) investigation funded under NASA's Prime contract no. NAS5-01072. We acknowledge H. Spence and G. Reeves for the ECT data, B. Blake for the MagEIS data and D. Baker for the REPT data. We are also

thankful to the Van Allen Probes, POES, GOES, Wind and OMNI teams for making their data publicly available.



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
