# Peer review of "Outer Van Allen belt trapped and precipitating electron flux responses to two interplanetary magnetic clouds of opposite polarity"

_Annales Geophysicae, 2020_

## Referee Comment (RC1) · Anonymous Referee #1 · 15 May 2020

General Comments:

This manuscript compares two events in which ICMEs impact Earth's magnetosphere and cause responses in both trapped and precipitating radiation belt electrons. They qualitatively compare and contrast the timing, characteristics, and magnetospheric response during these two events, one of which has magnetic cloud orientation and rotation of Bz North to South, and the other South to North. They describe in detail both events, but as currently presented, it is difficult to directly compare the events or attribute their differences specifically to the ICME characteristics. I recommend the following specific comments be addressed to help clarify the manuscript and its findings.

Specific Comments:

1. It is difficult to compare and contrast the two events directly as they are currently presented. Please consider incorporating some of the following potential suggestions to help the reader better identify the key take-aways from the two-event comparison:

- Combining Fig 1 and 3, 2 and 4 (as left and right panels, e.g.) would help the reader better look at the relative timing and magnitude of the magnetospheric response during these two events

- Including a summary table or figure in the discussion of the different key parameters that were investigated and their similarities/differences between events would also help, since there is a lot of detailed description and text in the discussion section to sift through.

2. Is there a reason the EMIC wave observations are taken from GOES rather than Van Allen Probes? Please include the MLT of both spacecraft during these observations, since this can have a large influence on which wave populations will be sampled. Additionally, summing the hiss and chorus wave power from the Van Allen Probes (as you've done for GOES Pc5 and EMIC measurements) in Fig 2 and 4 would make these plots easier to more quantitatively compare between the two events.

3. The plasmapause location might be more useful to show in the figures of radiation belt fluxes (5, 7) rather than solar wind data, so that the trapped fluxes inside and outside of the plasmasphere can be better identified in Van Allen Probes data.

4. POES data:

- was using the P6 channel considered, for comparison to the trapped MeV electron populations?

- line 501-503: Why is the precipitation enhancement here (as opposed to at other times) assumed to be due to a trapped flux enhancement? Some further justification of this is needed. This also raises the general question of how to interpret the POES

data at it is presented, if enhancements can be due just as easily to enhancements in trapped fluxes as enhancements in precipitation. While the caveats of the current technique for presenting the POES data (equation 1 of the manuscript) are nicely mentioned, it is not clear how much better they are than presenting just the 0deg telescope measurements for the purposes of this event comparison.

Technical Corrections:

1. please double check the color bar axis units (e.g. Fig 5) – it looks like the REPT >3.4MeV fluxes are larger than those in MagEIS >346 and >1079 keV channels.

2. line 381 – typo: "decreases" should be "decreased" or "decreasing"

3. line 387 – depletion at "high L-shell" is discussed, but then referenced as "low latitude" – shouldn't higher L shells map to higher latitudes?

4. line 400 – typo: "event 2 exhibits moderate level" -> "moderate levels"

---

## Referee Comment (RC2) · Anonymous Referee #2 · 16 May 2020

This manuscript compares observations from the inner magnetosphere during two different CME-driven storms. The storms are of a similar magnitude but driven by CMEs with opposite rotations of the Bz magnetic field component. The manuscript describes differences in the timing and features of the solar wind during the chosen storms and compares them with the observations of wave activity from RBSP and GOES, precipitating electron flux at POES, and source, seed, and radiation belt electron fluxes at RBSP. The manuscript concludes that the location and timing of the southward component of the magnetic field is a key factor in driving the differences in the timing of trapped and precipitating flux variations during CME-driven storms. The manuscript is very nicely written and provides new insight, but there are comments which should be

addressed prior to publication.

Main Comment: The manuscript currently needs to take better care addressing the effects of the local time of the spacecraft in their analysis. The RBSP spacecraft have quite different locations in apogee during the two events which are compared. During event 1 RBSP is primarily on the dayside. During event 2 it is near dusk. There is a local time dependency of chorus (e.g. Li et al., 2009; Meredith et al., 2012), source/seed electrons (e.g. Allison et al., 2017; Korth et al., 1999), and potentially a storm phase dependency on the local time of chorus/source electrons (Bingham et al., 2019). As such, one would not necessarily expect to observe the same timings and intensities of lower energy electron flux and chorus wave activity during each storm. While the manuscript has a thorough description of most of the timing of various features observed, this is one part that still needs to be better addressed.

Other comments: Chorus and hiss waves are not necessarily going to be the only waves present between 100–10000 Hz over an RBSP orbit. That is not to say that they will not be the dominant ones. Most of the features shown certainly look chorus-like and hiss-like. However, I think a little more care could be used either in describing caveats of the chorus/hiss observations as they are, or using the wave properties to provide greater certainty that the waves shown are in fact chorus/hiss.

While they will only be from a limited local time, including RBSP observations of the plasmapause location could provide useful context for the events and provide a comparison to the empirical model currently used. Additionally, over plotting the empirical and/or observed plasmapause location on the RBSP electron fluxes would help the reader.

Lines 189-190. "The use of two RBSP satellites over a period of multiple days meant that all MLT were encompassed". RBSP is not able to cover all MLT for all L-shells shown during each event.

Line 297-298. "Dst begins to decrease rather steadily soon after the ejecta leading

edge arrives at Earth, reaching a minimum value of $-102$ nT on June 29, 2013, 07:00 UT." How much of the initial decrease in Dst is due to passing of the sheath region and the end of the sudden storm commencement? After an initial decrease, Dst seems to be at a rather constant value, which is pretty comparable to the prestorm value, for the first $\sim$8 hours of 28/06/13.

Line 381. Typo: "The source population flux is strongly decreases towards the time of the ejecta trailing edge"

Figures 1-4. Minor tickmarks on the x-axis every few hours would be helpful to the reader.

Similarly, many of the line flux plots with a log y-axis could use more tickmarks on the y-axis for reference.

Allison et al. (2017). The magnetic local time distribution of energetic electrons in the radiation belt region. JGR. doi: 10.1002/2017JA024084

Bingham et al. (2019). The Storm Time Development of Source Electrons and Chorus Wave Activity During CME and CIR Driven Storms. JGR. DOI:10.1029/2019JA026689

Korth et al. (1999). Plasma sheet access to geosynchronous orbit. JGR. DOI:10.1029/1999JA900292 Li et al. (2009). Global distribution of whistler-mode chorus waves observed on the THEMIS spacecraft. GRL. DOI:10.1029/2009GL037595

Meredith et al. (2012). Global model of lower band and upper band chorus from multiple satellite observations. JGR. DOI:10.1029/2012JA017978

---

## Author Comment (AC1) · 23 Jun 2020

General referee comment: This manuscript compares two events in which ICMEs impact Earth's magnetosphere and cause responses in both trapped and precipitating radiation belt electrons. They qualitatively compare and contrast the timing, characteristics, and magnetospheric response during these two events, one of which has magnetic cloud orientation and rotation of Bz North to South, and the other South to North. They describe in detail both events, but as currently presented, it is difficult to directly compare the events or attribute their differences specifically to the ICME characteristics. I recommend the following specific comments be addressed to help clarify

the manuscript and its findings.

Response: We thank the referee for the careful reading of our manuscript and their constructive comments. We have revised the paper accordingly. Please find our responses below.

Referee comment: It is difficult to compare and contrast the two events directly as they are currently presented. Please consider incorporating some of the following potential suggestions to help the reader better identify the key take-aways from the two-event comparison: - Combining Fig 1 and 3, 2 and 4 (as left and right panels, e.g.) would help the reader better look at the relative timing and magnitude of the magnetospheric response during these two events

Response: Figures 1 and 3, and 2 and 4 have been combined as suggested. These figures are attached below

Referee comment: Including a summary table or figure in the discussion of the different key parameters that were investigated and their similarities/differences between events would also help, since there is a lot of detailed description and text in the discussion section to sift through.

Response: A color-coded summary table of selected solar wind conditions, wave activity, and electron flux response during the sheath, ejecta, and recovery phases of the two events has been added. This table is on page 22 of the revised manuscript.

Referee comment: Is there a reason the EMIC wave observations are taken from GOES rather than Van Allen Probes? Please include the MLT of both spacecraft during these observations, since this can have a large influence on which wave populations will be sampled.

Response: MLT information for the Van Allen Probes have been added to the Wave Activity figure (attached) to show when they are on the dayside and nightside of the Earth. In this figure, a dashed line represents that the satellite is on the nightside while

a solid line represents dayside satellite location. The following text has been added to the Data and Methods section (line 160 of the revised manuscript) to explain the use of GOES data for the EMIC waves: "RBSP data can be used to calculate local ULF and EMIC wave powers on shorter timescales, but these data are not ideal for the analysis of ULF or EMIC waves over the course of an event. The RBSP travel rapidly over a highly elliptical orbit, so sample a range of plasma environments from different regions of the inner magnetosphere over the course of a half orbit. The ULF waves are global and EMIC waves are long lasting, meaning that they are poorly observed over long time periods by the RBSP. By comparison, the GOES satellites are better suited to ULF and EMIC observations over a longer time period due to their fixed orbit and longer period."

Referee comment: Additionally, summing the hiss and chorus wave power from the Van Allen Probes (as you've done for GOES Pc5 and EMIC measurements) in Fig 2 and 4 would make these plots easier to more quantitatively compare between the two events.

Response: The hiss and chorus wave power has been summed as suggested. This can be seen in the attached Wave Conditions figure.

Referee comment: The plasmapause location might be more useful to show in the figures of radiation belt fluxes (5, 7) rather than solar wind data, so that the trapped fluxes inside and outside of the plasmasphere can be better identified in Van Allen Probes data.

Response: We have plotted an overlay of the magnetopause and plasmapause position on the colour maps of precipitating and trapped electron flux, in addition to showing the magnetopause and plasmapause location in the Solar Wind Conditions figure. The overlay of the magnetopause is orange and the plasmapause overlay is magenta. The colour maps with these overlays have been attached.

Referee comment: Was using the P6 channel considered, for comparison to the

trapped MeV electron populations?

Response: No. Channel P6 measures protons but has significant contamination from relativistic electrons. Measurements from this channel have been previously used in qualitative analysis of relativistic electron precipitation (Peck et. al, 2015, 10.1002/2014JA020817) by cross-referencing the P6 measurements with other proton measurements to isolate the electron contamination (Rodger et al., 2010, 10.1029/2008JA014023). However, the P6 channel is not very sensitive and the precipitating fluxes of relativistic electrons is generally quite low due to the low proportion of relativistic electrons in the outer radiation belt to electrons with tens or hundreds of keV. This means that the P6 channel does not always provide reliable data for precipitating electrons (Yando et. al., 2011, 10.1029/2011JA016671), and can therefore be difficult to interpret for use in electron precipitation studies. As a result of these complications, we elected to focus on direct measurements of electron flux in the lower energy channels rather than attempt to incorporate indirect measurements of high energy electrons.

Referee comment: line 501-503: Why is the precipitation enhancement here (as opposed to at other times) assumed to be due to a trapped flux enhancement? Some further justification of this is needed.

Response: Upon further consideration, we have changed our explanation of this precipitation enhancement. This text: "The observed precipitation flux enhancement occurring prior to the trapped flux enhancement is likely not due to a true increase in precipitating flux at this time, but rather due to the $90^\circ$ POES telescope measuring the trapped flux enhancement before it was measured by RBSP." ...has been changed to: "The precipitating flux enhancement preceding the trapped flux enhancement may be due to increased chorus and EMIC wave activity at this time causing greater scattering of the existing lower energy populations into the bounce-loss cone. A different mechanism may have then caused the trapped flux enhancement at a slightly later time."

Referee comment: This also raises the general question of how to interpret the POES data as it is presented, if enhancements can be due just as easily to enhancements in trapped fluxes as enhancements in precipitation. While the caveats of the current technique for presenting the POES data (equation 1 of the manuscript) are nicely mentioned, it is not clear how much better they are than presenting just the 0deg telescope measurements for the purposes of this event comparison.

Response: The following text has been added to the Data and Methods section: "This means that the POES $0°$ detectors do not resolve fluxes near the edge of the loss cone in the case of partially filled loss cones. Therefore, the $0°$ detector chronically underestimates fluxes in the loss cone... Despite these drawbacks, we expect this approach for precipitating flux will be superior to the direct $0°$ telescope measurements, because it attempts to provide a more accurate estimate of the loss cone fluxes for the latitudes evaluated in this study."

Technical Corrections: 1. please double check the color bar axis units (e.g. Fig 5) – it looks like the REPT >3.4MeV fluxes are larger than those in MagEIS >346 and >1079 keV channels. This has been corrected. REPT units were per MeV while MagEIS were per keV. The data has been adjusted so all fluxes are per keV 2. line 381 – typo: "decreases" should be "decreased" or "decreasing" Corrected: decreases -> decreasing 3. line 387 – depletion at "high L-shell" is discussed, but then referenced as "low latitude" – shouldn't higher L shells map to higher latitudes? Corrected: low latitude -> high latitude 4. line 400 – typo: "event 2 exhibits moderate level" -> "moderate levels" Corrected: level -> levels

**Solar Wind Conditions**

**Event 1 : 12/30, 2015 — 01/02, 2016**

1a) Magnetic Field Magnitude

1b) Magnetic Field Components

1c) Solar Wind Velocity

1d) Solar Wind Dynamic Pressure

1e) Magnetopause Position

1f) Geomagnetic Indices

1g) Plasmapause Location

**Event 2 : 06/26, 2013 — 06/30, 2013**

2a) Magnetic Field Magnitude

2b) Magnetic Field Components

2c) Solar Wind Velocity

2d) Solar Wind Dynamic Pressure

2e) Magnetopause Position

2f) Geomagnetic Indices

2g) Plasmapause Location

**Fig. 1.**

**Wave Conditions**

**Event 1 : 12/30, 2015 − 01/02, 2016**

1a) RBSP-A

1b) RBSP-B

1c) GOES-13

1d) GOES-15

**Event 2 : 06/26, 2013 − 06/30, 2013**

2a) RBSP-A

2b) RBSP-B

2c) GOES-13

2d) GOES-15

Legend (a, b): Lower band, Upper band, Hiss

Legend (c, d): Pc5, EMIC

Fig. 2.

[Figure]

[Figure]

**Fig. 3.**

[Figure]

[Figure]

**Fig. 4.**

[Figure]

---

## Author Comment (AC2) · 23 Jun 2020

General Referee Comment: This manuscript compares observations from the inner magnetosphere during two different CME-driven storms. The storms are of a similar magnitude but driven by CMEs with opposite rotations of the Bz magnetic field component. The manuscript describes differences in the timing and features of the solar wind during the chosen storms and compares them with the observations of wave activity from RBSP and GOES, precipitating electron flux at POES, and source, seed, and radiation belt electron fluxes at RBSP. The manuscript concludes that the location and timing of the southward component of the magnetic field is a key factor in driving the

differences in the timing of trapped and precipitating flux variations during CME-driven storms. The manuscript is very nicely written and provides new insight, but there are comments which should be addressed prior to publication.

Response: We thank the referee for the careful reading of our manuscript and their constructive comments. We have revised the paper accordingly. Please find our responses below.

Main Referee Comment: The manuscript currently needs to take better care addressing the effects of the local time of the spacecraft in their analysis. The RBSP spacecraft have quite different locations in apogee during the two events which are compared. During event 1 RBSP is primarily on the dayside. During event 2 it is near dusk. There is a local time dependency of chorus (e.g. Li et al., 2009; Meredith et al., 2012), source/seed electrons (e.g. Allison et al., 2017; Korth et al., 1999), and potentially a storm phase dependency on the local time of chorus/source electrons (Bingham et al., 2019). As such, one would not necessarily expect to observe the same timings and intensities of lower energy electron flux and chorus wave activity during each storm. While the manuscript has a thorough description of most of the timing of various features observed, this is one part that still needs to be better addressed.

Response: Additional information on the effects of MLT on electron flux response and wave activity has been incorporated throughout the manuscript. A description of the magnetic local time of the Van Allen Probes in these two events has also been incorporated into the text. MLT information for the RBSP have been added to the Wave Activity figure (attached) to show when they are on the dayside and nightside of the Earth. In this figure, a solid line represents that the RBSP is on the dayside of the Earth while a dashed line represents that the satellite is on the nightside.

Referee comment: Chorus and hiss waves are not necessarily going to be the only waves present between 100–10000 Hz over an RBSP orbit. That is not to say that they will not be the dominant ones. Most of the features shown certainly look chorus-like

and hiss-like. However, I think a little more care could be used either in describing caveats of the chorus/hiss observations as they are, or using the wave properties to provide greater certainty that the waves shown are in fact chorus/hiss.

Response: The following text has been added to the Data and Methods section: "We have elected to use only the wave frequency and location inside or outside the plasmasphere to categorise these waves as either chorus waves or plasmaspheric hiss for simplicity."

Referee comment: While they will only be from a limited local time, including RBSP observations of the plasmapause location could provide useful context for the events and provide a comparison to the empirical model currently used.

Response: We agree that inclusion of RBSP measurements of plasmapause location would be useful. However, we have elected not to do this. There is some arbitrariness involved with the selection of the electron density threshold for the plasmapause location when using RBSP data (e.g. Goldstein et. al., 10.1002/2014JA020252, 2014), and there is also limited electron density data for the dates in this study. We think that the plasmapause location from the empirical model is sufficient for the purposes of the study.

Referee comment: Additionally, over plotting the empirical and/or observed plasmapause location on the RBSP electron fluxes would help the reader.

Response: We have plotted an overlay of the plasmapause and magnetopause position on the colour maps of precipitating and trapped electron flux, in addition to showing the magnetopause and plasmapause location in the Solar Wind Conditions figure. The plasmapause overlay is magenta and the magnetopause position is shown in orange. This can be seen in the attached figures of electron fluxes during the two events.

Referee comment: Lines 189-190. "The use of two RBSP satellites over a period of multiple days meant that all MLT were encompassed". RBSP is not able to cover all

MLT for all L-shells shown during each event.

Response: The text has been edited to clarify that all MLT were covered at some point in the evaluated time periods, not that every MLT was covered at every L-shell

Referee comment: Line 297-298. "Dst begins to decrease rather steadily soon after the ejecta leading edge arrives at Earth, reaching a minimum value of -102 nT on June 29, 2013, 07:00 UT." How much of the initial decrease in Dst is due to passing of the sheath region and the end of the sudden storm commencement? After an initial decrease, Dst seems to be at a rather constant value, which is pretty comparable to the prestorm value, for the first 8 hours of 28/06/13.

Response: Changed "soon" to "approximately eight hours" and added the sentence "There is a minor decrease in Dst at the ejecta leading edge, but it remains close to pre-event levels." to better explain the evolution of the Dst index throughout the sheath region and early portion of the ejecta

Referee comment: Line 381. Typo: "The source population flux is strongly decreases towards the time of the ejecta trailing edge"

Response: Corrected: decreases -> decreasing

Referee comment: Figures 1-4. Minor tickmarks on the x-axis every few hours would be helpful to the reader.

Response: Additional minor tickmarks have been added to the x-axis in all figures

Referee comment: Similarly, many of the line flux plots with a log y-axis could use more tickmarks on the y-axis for reference.

Response: Minor y-axis tickmarks have been added to the Median Flux plots

Suggested citations from referee: Allison et al. (2017). The magnetic local time distribution of energetic electrons in the radiation belt region. JGR. doi: 10.1002/2017JA024084

Bingham et al. (2019). The Storm Time Development of Source Electrons and Chorus Wave Activity During CME and CIR Driven Storms. JGR. DOI:10.1029/2019JA026689

Korth et al. (1999). Plasma sheet access to geosynchronous orbit. JGR. DOI:10.1029/1999JA900292 Li et al. (2009). Global distribution of whistler-mode chorus waves observed on the THEMIS spacecraft. GRL. DOI:10.1029/2009GL037595

Meredith et al. (2012). Global model of lower band and upper band chorus from multiple satellite observations. JGR. DOI:10.1029/2012JA017978

Response: The suggested citations have been incorporated into the Introduction section. A paragraph has been added on MLT dependence of trapped electron fluxes at the energies evaluated in this study. Further information on the impact of MLT and geomagnetic activity on chorus wave activity and distribution has also been added to the paragraph on magnetospheric wave activity.

[Figure]

**Wave Conditions**

1a) Event 1 : 12/30, 2015 − 01/02, 2016
RBSP-A

1b) RBSP-B

1c) GOES-13

1d) GOES-15

2a) Event 2 : 06/26, 2013 − 06/30, 2013
RBSP-A

2b) RBSP-B

2c) GOES-13

2d) GOES-15

**Fig. 1.**

[Figure]

Fig. 2.

[Figure]

[Figure]

**Fig. 3.**

---

## Author Response (AR1)

General Comments:
This manuscript compares two events in which ICMEs impact Earth's magnetosphere and cause responses in both trapped and precipitating radiation belt electrons. They qualitatively compare and contrast the timing, characteristics, and magnetospheric response during these two events, one of which has magnetic cloud orientation and rotation of Bz North to South, and the other South to North. They describe in detail both events, but as currently presented, it is difficult to directly compare the events or attribute their differences specifically to the ICME characteristics. I recommend the following specific comments be addressed to help clarify the manuscript and its findings.

**We thank the referee for the careful reading of our manuscript and his/her constructive comments. We have revised the paper accordingly. Please find our responses below.**

Specific Comments:
1. It is difficult to compare and contrast the two events directly as they are currently presented. Please consider incorporating some of the following potential suggestions to help the reader better identify the key take-aways from the two-event comparison:
- Combining Fig 1 and 3, 2 and 4 (as left and right panels, e.g.) would help the reader better look at the relative timing and magnitude of the magnetospheric response during these two events
**Figures 1 and 3, and 2 and 4 have been combined as suggested.**

- Including a summary table or figure in the discussion of the different key parameters that were investigated and their similarities/differences between events would also help, since there is a lot of detailed description and text in the discussion section to sift through.
**A color-coded summary table of selected solar wind conditions, wave activity, and electron flux response during the sheath, ejecta, and recovery phases of the two events has been added.**

2. Is there a reason the EMIC wave observations are taken from GOES rather than Van Allen Probes? Please include the MLT of both spacecraft during these observations, since this can have a large influence on which wave populations will be sampled.
**MLT information for the Van Allen Probes have been added to the Wave Activity figure to show when they are on the dayside and nightside of the Earth.**

**The following text has been added to the Data and Methods section to explain the use of GOES data for the EMIC waves:**
**"RBSP data can be used to calculate local ULF and EMIC wave powers on shorter timescales, but these data are not ideal for the analysis of ULF or EMIC waves over the**

course of an event. The global nature of these waves mean that they are poorly observed over long time periods by the RBSP, as these satellites rapidly move through the different regions of the inner magnetosphere, sampling a range of plasma environments over the course of a half-orbit. By comparison, the GOES satellites are better suited to ULF and EMIC observations over a longer time period as they remain at the same distance for the duration of the analysed events."**

Additionally, summing the hiss and chorus wave power from the Van Allen Probes (as you've done for GOES Pc5 and EMIC measurements) in Fig 2 and 4 would make these plots easier to more quantitatively compare between the two events.
**The hiss and chorus wave power has been summed as suggested.**

3. The plasmapause location might be more useful to show in the figures of radiation belt fluxes (5, 7) rather than solar wind data, so that the trapped fluxes inside and outside of the plasmasphere can be better identified in Van Allen Probes data.
**We have plotted an overlay of the magnetopause and plasmapause position on the colour maps of precipitating and trapped electron flux, in addition to showing the magnetopause and plasmapause location in the Solar Wind Conditions figure.**

4. POES data:
- was using the P6 channel considered, for comparison to the trapped MeV electron populations?
**No. Channel P6 measures protons but has significant contamination from relativistic electrons. Measurements from this channel have been previously used in qualitative analysis of relativistic electron precipitation (Peck et. al, 2015, 10.1002/2014JA020817) by cross-referencing the P6 measurements with other proton measurements to isolate the electron contamination (Rodger et al., 2010, 10.1029/2008JA014023). According to Rodger et al., "there is an energy-dependent time delay observed in the POES/SEM-2 observations, with an almost one-week delay between the >30 keV electron enhancement and the P6omni relativistic electron enhancement". Since each of the evaluated time periods were less than a week, the relativistic electron flux enhancements / depletions caused by the ICME would likely not be observed in the P6 channel. Therefore, we elected to focus on direct measurements of electron flux in the lower energy channels rather than attempt to incorporate indirect measurements of high energy electrons.**

- line 501-503: Why is the precipitation enhancement here (as opposed to at other times) assumed to be due to a trapped flux enhancement? Some further justification of this is needed.
**Upon further consideration, we have changed our explanation of this precipitation enhancement.**
**This text:**
**"The observed precipitation flux enhancement occurring prior to the trapped flux enhancement is likely not due to a true increase in precipitating flux at this time, but**

**rather due to the 90º POES telescope measuring the trapped flux enhancement before it was measured by RBSP."**
...has been changed to:
**"The precipitating flux enhancement preceding the trapped flux enhancement may be due to increased chorus and EMIC wave activity at this time causing greater scattering of the existing lower energy populations into the bounce-loss cone. A different mechanism may have then caused the trapped flux enhancement at a slightly later time."**

This also raises the general question of how to interpret the POES data as it is presented, if enhancements can be due just as easily to enhancements in trapped fluxes as enhancements in precipitation. While the caveats of the current technique for presenting the POES data (equation 1 of the manuscript) are nicely mentioned, it is not clear how much better they are than presenting just the 0deg telescope measurements for the purposes of this event comparison.
**The following text has been added to the Data and Methods section:**
**"This means that the POES 0º detectors do not resolve fluxes near the edge of the loss cone in the case of partially filled loss cones. Therefore, the 0º detector chronically underestimates fluxes in the loss cone… Despite [the drawbacks of Eq. 1], this approach for precipitating flux is still superior to the direct 0º telescope measurements because it provides a more accurate estimate of the total fluxes in the loss cone at the latitudes evaluated in this study."**

Technical Corrections:
1. please double check the color bar axis units (e.g. Fig 5) – it looks like the REPT >3.4MeV fluxes are larger than those in MagEIS >346 and >1079 keV channels.
**This has been corrected. REPT units were per MeV while MagEIS were per keV. The data has been adjusted so all fluxes are per keV**
2. line 381 – typo: "decreases" should be "decreased" or "decreasing"
**Corrected: decreases -> decreasing**
3. line 387 – depletion at "high L-shell" is discussed, but then referenced as "low latitude" – shouldn't higher L shells map to higher latitudes?
**Corrected: low latitude -> high latitude**
4. line 400 – typo: "event 2 exhibits moderate level" -> "moderate levels"
**Corrected: level -> levels**

**Anonymous Referee #2**

This manuscript compares observations from the inner magnetosphere during two different CME-driven storms. The storms are of a similar magnitude but driven by CMEs
with opposite rotations of the Bz magnetic field component. The manuscript describes
differences in the timing and features of the solar wind during the chosen storms and

compares them with the observations of wave activity from RBSP and GOES, precipitating electron flux at POES, and source, seed, and radiation belt electron fluxes at RBSP. The manuscript concludes that the location and timing of the southward component of the magnetic field is a key factor in driving the differences in the timing of trapped and precipitating flux variations during CME-driven storms. The manuscript is very nicely written and provides new insight, but there are comments which should be addressed prior to publication.

**We thank the referee for the careful reading of our manuscript and his/her constructive comments. We have revised the paper accordingly. Please find our responses below.**

Main Comment:
The manuscript currently needs to take better care addressing the effects of the local time of the spacecraft in their analysis. The RBSP spacecraft have quite different locations in apogee during the two events which are compared. During event 1 RBSP is primarily on the dayside. During event 2 it is near dusk. There is a local time dependency of chorus (e.g. Li et al., 2009; Meredith et al., 2012), source/seed electrons (e.g. Allison et al., 2017; Korth et al., 1999), and potentially a storm phase dependency on the local time of chorus/source electrons (Bingham et al., 2019). As such, one would not necessarily expect to observe the same timings and intensities of lower energy electron flux and chorus wave activity during each storm. While the manuscript has a thorough description of most of the timing of various features observed, this is one part that still needs to be better addressed.

**Additional information on the effects of MLT on electron flux response and wave activity has been incorporated into the manuscript. Information on local time of the RBSP in these two events has also been added. See the responses below and the responses to Referee 1's comments for details.**

Other comments:
Chorus and hiss waves are not necessarily going to be the only waves present between 100–10000 Hz over an RBSP orbit. That is not to say that they will not be the dominant ones. Most of the features shown certainly look chorus-like and hiss-like. However, I think a little more care could be used either in describing caveats of the chorus/hiss observations as they are, or using the wave properties to provide greater certainty that the waves shown are in fact chorus/hiss.
**The following text has been added to the Data and Methods section: "We have elected to use only the wave frequency and location inside or outside the plasmasphere to categorise these waves as either chorus waves or plasmaspheric hiss for simplicity."**

While they will only be from a limited local time, including RBSP observations of the plasmapause location could provide useful context for the events and provide a comparison

to the empirical model currently used.

**We agree that inclusion of RBSP measurements of plasmapause location would be useful. However, we have elected not to do this. There are significant uncertainties involved with calculating plasmapause location from RBSP electron density data, and there is also limited electron density data for the dates in this study. We think that the plasmapause location from the empirical model is sufficient for the purposes of the study.**

Additionally, over plotting the empirical and/or observed plasmapause location on the RBSP electron fluxes would help the reader.

**We have plotted an overlay of the plasmapause and magnetopause position on the colour maps of precipitating and trapped electron flux, in addition to showing the magnetopause and plasmapause location in the Solar Wind Conditions figure.**

Lines 189-190. "The use of two RBSP satellites over a period of multiple days meant that all MLT were encompassed". RBSP is not able to cover all MLT for all L-shells shown during each event.

**The text has been edited to clarify that all MLT were covered at some point in the evaluated time periods, not that every MLT was covered at every L-shell**

Line 297-298. "Dst begins to decrease rather steadily soon after the ejecta leading edge arrives at Earth, reaching a minimum value of 102 nT on June 29, 2013, 07:00 UT." How much of the initial decrease in Dst is due to passing of the sheath region and the end of the sudden storm commencement? After an initial decrease, Dst seems to be at a rather constant value, which is pretty comparable to the prestorm value, for the first 8 hours of 28/06/13.

**Changed "soon" to "approximately eight hours" and added the sentence "There is a minor decrease in Dst at the ejecta leading edge, but it remains close to pre-event levels." to better explain the evolution of the Dst index throughout the sheath region and early portion of the ejecta**

Line 381. Typo: "The source population flux is strongly decreases towards the time of the ejecta trailing edge"

**Corrected: decreases -> decreasing**

Figures 1-4. Minor tickmarks on the x-axis every few hours would be helpful to the reader.
**Additional minor tickmarks have been added to the x-axis in all figures**

Similarly, many of the line flux plots with a log y-axis could use more tickmarks on the y-axis for reference.
**Minor y-axis tickmarks have been added to the Median Flux plots**

- Allison et al. (2017). The magnetic local time distribution of energetic electrons in the radiation belt region. JGR. doi: 10.1002/2017JA024084
- Bingham et al. (2019). The Storm Time Development of Source Electrons and Chorus Wave Activity During CME and CIR Driven Storms. JGR. DOI:10.1029/2019JA026689
- Korth et al. (1999). Plasma sheet access to geosynchronous orbit. JGR. DOI:10.1029/1999JA900292
- Li et al. (2009). Global distribution of whistler-mode chorus waves observed on the THEMIS spacecraft. GRL. DOI:10.1029/2009GL037595
- Meredith et al. (2012). Global model of lower band and upper band chorus from multiple satellite observations. JGR. DOI:10.1029/2012JA017978

**The suggested citations have been incorporated into the Introduction section. A paragraph has been added on MLT dependence of trapped electron fluxes at the energies evaluated in this study. Further information on the impact of MLT and geomagnetic activity on chorus wave activity and distribution has also been added to the paragraph on magnetospheric wave activity.**

[revised manuscript text omitted]